# Model certainty in cellular network-driven processes with missing data

**Michael W. Irvin[1,2], Arvind Ramanathan[2], Carlos F. Lopez[1,3¤]***

**1** Department of Biochemistry, Vanderbilt University, Nashville, Tennessee, United States of America,
**2** Data Science and Learning Division, Argonne National Laboratory, Lemont, Illinois, United States of
America, **3** Department of Biomedical Informatics, Vanderbilt University Medical Center, Nashville
Tennessee, United States of America

¤ Current address: Multiscale Modeling Group, Altos Labs, Redwood City, California, United States of
America
* clopez@altoslabs.com

pcbi.1011004

Institutes of Health, UNITED STATES

**Data Availability Statement:** The computer code
used to perform the analyses in this article is
available in the following GitHub repositories:
https://github.com/LoLab-MSM/Opt2Q and https://
github.com/LoLab-MSM/aEARM_cell_viability_

## Abstract

Mathematical models are often used to explore network-driven cellular processes from a
systems perspective. However, a dearth of quantitative data suitable for model calibration
leads to models with parameter unidentifiability and questionable predictive power. Here we
introduce a combined Bayesian and Machine Learning Measurement Model approach to
explore how quantitative and non-quantitative data constrain models of apoptosis execution
within a missing data context. We find model prediction accuracy and certainty strongly
depend on rigorous data-driven formulations of the measurement, and the size and make-
up of the datasets. For instance, two orders of magnitude more ordinal (e.g., immunoblot)
data are necessary to achieve accuracy comparable to quantitative (e.g., fluorescence)
data for calibration of an apoptosis execution model. Notably, ordinal and nominal (e.g., cell
fate observations) non-quantitative data synergize to reduce model uncertainty and improve
accuracy. Finally, we demonstrate the potential of a data-driven Measurement Model
approach to identify model features that could lead to informative experimental measure-
ments and improve model predictive power.

## Author summary

Mathematical models used to explore network-driven cellular processes from a systems
perspective face a challenge in the dearth of quantitative data required for model calibra-
tion. We address this challenge with a combined Bayesian and Machine Learning mea-
surement model approach to explore how quantitative and many non-quantitative data
constrain models of apoptosis execution. We find model prediction accuracy and cer-
tainty depend on rigorous data-driven formulations of the measurement, and the size and
make-up of the datasets. We also find different non-quantitative dataset can be combined
to synergistically better support model calibration. Finally, we demonstrate the use of a
data-driven Measurement Model approach to identify model features that could lead to
informative experimental measurements and improve model predictive power.

calibration. Data associated with this work can be loaded from https://doi.org/10.5281/zenodo.7007655.

**Funding:** This work was supported by the following funding sources: MWI was supported by the National Institutes of Health (NIH)[T32-GM139800]; CFL was supported by the National Science Foundation (NSF) CAREER Award [MCB 1942255]; and the National Institutes of Health (NIH) [U54-CA217450 and U01-CA215845]. The funders had no role in study design, data collection and analysis, decision to publish, or preparation of the manuscript.

**Competing interests:** The authors have declared that no competing interests exist.

## Introduction

The combination of systems approaches and quantitative data, promised a novel understanding of cellular mechanisms that could spur science-driven innovation in biology and medicine–as has happened in physics, chemistry, and engineering [1–3]. However, despite massive research efforts and data accumulation, a systems level understanding of cellular regulation, signaling and many other processes remains rudimentary. The quantitative and systems biology fields continue to employ strategies from physics and engineering to construct models of biological mechanism from first principles [4,5]. Yet these strategies are incompatible with the types of measurements and observations that predominate biological investigations. Experiments investigating cell fate outcomes (such as apoptosis) produce cellular phenotype and other categorical observations, which are hard to define in terms of variables encoded in mathematical mechanistic models of biological processes [6]. Therefore, the connection of mechanistic models to corresponding biological measurements is subject to practitioner interpretation. As a result, vast amounts of existing nonquantitative data in cell biology have led to mechanistic formulations based on simple inference and informal reasoning. The problem is exacerbated by additional confounding factors such as noise, complexity, and hierarchical organization of organisms, which limits how one can experimentally perturb and measure biological systems [7,8]. This relative dearth of quantitative data in turn has led to mechanistic models with poorly constrained parameters and unreliable predictions. Unfortunately, both non-quantitative and quantitative data, collected in an unplanned manner, result in missed opportunities to quantitatively explain complex cellular mechanisms [9]. The present work is motivated by a need to rigorously define the relationship between measurements and biological knowledge extracted from experimental data.

The data-to-knowledge problem in biology has prompted researchers to incorporate non-quantitative data as a complement or substitute for quantitative data in the development of mechanistic models [10–13]. The traditional workflow employed to train mechanistic models to data comprises mechanistic models and experimental measurements linked through an objective function that is optimized via a model calibration method (Fig 1A) [14, 15]. The objective function typically *implies* a simple equivalency (or linear) relationship between the mechanistic model and corresponding experimental measurements. However, this relation can preclude incorporation of common measurements in biology, which do not have a direct relation to a given mathematical model. Recent innovations have adapted traditional mechanistic model calibration strategies to incorporate nonquantitative data into mechanistic models and have revealed their intrinsic value in mechanistic hypothesis exploration. For example, pioneering work by Pargett and co-workers employed optimal scaling and multi-objective optimization for training mechanistic models to large ordinal datasets [10]. Schmiester et al. incorporated this strategy into PyPESTO, a model parameter estimation framework [11]. The work by Pargett et al. and Schmiester et al. imposes discrete boundaries on the mechanistic model to reflect discrete ordinal values in the data. This approach adjusts the placement of these discrete boundaries to optimize the model's fit to data. Unfortunately, the discrete boundaries in this approach limit the ability to apply Bayesian methods and integrate multiple datatypes for training and uncertainty estimation of mechanistic models. More recently, Mitra et al. applied predefined constraint-based models of categorical data and modified their approach to allow definition of a likelihood function within a Bayesian formalism [12, 13]. However, the *ad hoc* nature of their constraint models can still lead to biasing assumptions. While recent advances focus on incorporating categorical data into the development of mechanistic models, traditional model calibration methods also preclude use of common *quantitative* measurements in biology. For example, ubiquitous measurements such as cell viability (or

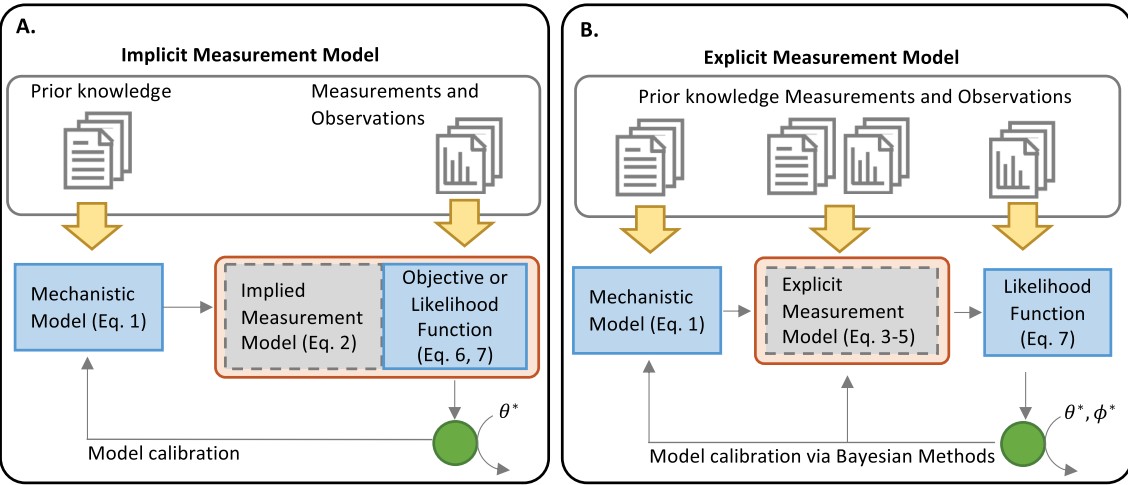

Fig 1. Objective functions and the role of a measurement model. Mechanistic models of biological processes are typically encoded as systems of (ordinary) differential equations (Eq 1). Model calibration relies on an objective function (Eq 6)—or in a Bayesian setting, a likelihood function (Eq 7)—quantifies the degree of dissimilarity or similarity between model variables and corresponding measurements. Note, the objective or likelihood function uses an *implied* measurement model (Eq 6) which converts modeled variables $x(t)$ to a quantity $y(t_i, \theta)$ that can be compared to data $\hat{y}(t_i)$. In physics and engineering, where measurements are typically quantitative,

this *implicit* measurement model suffices. For nonquantitative measurements and observations, the measurement model must be defined explicitly in consideration of the nonquantitative measurements' properties.

fractional cell fate measurements) lack a definitive connection to variables encoded in models of intracellular mechanism. Schmiester et al. introduces models of fractional cell fate that impose a functional relationship between the particular variables in the mechanistic model and corresponding fractional cell fate measurements [16]. While amenable to Bayesian methods, their approach requires *ad hoc* assumptions about which features describe cell fate and how those features map to values of fractional cell fate [16]. These advances all have in common *explicitly* defined functions relating the mechanistic model to corresponding experimental measurements. These explicitly defined functions in turn highlight a bias-variance tradeoff inherent in *explicitly* defined functional relationships between mechanistic models and corresponding measurements. [17]. Given the limited application of Bayesian methods and biases introduced by *ad hoc* assumptions, however, the field still has a limited understanding of the contribution of nonquantitative and quantitative data to mechanistic knowledge in biological systems.

In this work, we tackle the data-to-knowledge challenge by detailing the application of *measurement models* in systems biology approaches. Measurement models aim to rigorously define measurements and observations in terms of an underlying mechanism (Fig 1B) [18]. This definition entails formulation of a function that maps variables encoded in a mechanistic model to values in the observed data. We note measurement models may comprise *any* functional relationship between the mechanistic model and accompanying general measurements and observations. Our approach departs from previous work in that it uses Machine Learning regression and classification methods to define explicit measurement models that 1) permit application of Bayesian methods, 2) flexibly encode knowledge and assumptions about the measurements as with free parameters and associated priors and 3) *learn* phenomenological connections between measurements and underlying mechanisms. We use a probabilistic definition of categorical measurements that lends itself to Bayesian methods and can therefore provide an unbiased evaluation of the predictive power of models trained to nonquantitative data. In what follows, we present our findings about common types of biological measurements, followed by a presentation of our methodology, and an examination of how nonquantitative and quantitative measurements affect mechanistic models' *a posteriori* predictive power. We employ a mechanistic model of apoptosis execution, in which all parameters are identifiable, to demonstrate how the amount, and type, of data applied to a mechanistic model can affect its predictive power. Apoptosis is a well-established and studied cellular signaling pathway involved in many biological processes in health, disease, and development [19]. Importantly for our goals, quantitative and nonquantitative empirical data have been made available for measurements of apoptosis execution [20, 21].We investigate how *ad hoc* assumptions in a measurement model can lead to spurious results, and further show how recasting these assumptions as flexible *a priori* parameterizations of the measurement model can be reexamined within a Bayesian, data-driven context. Subsequently, we further demonstrate the potential for Machine Learning measurement model formulations to learn phenomenological links between features (e.g., dynamic biomolecular signals) that predict and/or drive an emergent biological phenotype (e.g., apoptosis or survival). Finally, we validate the measurement model concept by calibrating our apoptosis model to published cell viability data, and we demonstrate that the measurement model learns how the variables in the mechanistic model map to values of fractional cell fate the context of data that has no known functional relation to mechanistic model quantities.

## Results

### Contributions of different data types to mechanistic models of Apoptosis

We first explored how experimental measurements aid the development of mathematical models (Eq 1, Fig 1) of cellular processes. Mechanistic models typically employ mass-actions formalisms comprised of reaction rates and chemical species concentrations to represent known and hypothesized networks of biochemical reactions. Direct quantitative measurement of all chemical reactions and species would provide needed model parameters to carry out simulations and *in silico* experiments. However, these measurements are typically not available and likely untenable for real systems, thus necessitating indirect measurements to infer model parameter values using estimators such as objective (Eq 6 and Fig 1) or likelihood functions (Eq 7 and Fig 1). When these functions are optimized, the resulting mathematical model can provide valuable new predictions and insights about the cellular process.

Experimental measurements occur in an unplanned manner largely to answer a series of individual biological questions, rather than intentional contributions to a mechanistic model. Measurements are processes that assign an empirical value to the *measurand* (i.e., property being investigated). Salient features of the measurement process dictate how we define, interpret, and apply the resulting data (See Fig 2 for further details). Biological measurements could be classified into two broad types: categorical and continuous. In this work we consider nominal and ordinal categorical measurements. *Nominal* data define distinct values (e.g., apoptosis vs. survival), but lack ordering or defined intervals between them (Eq 8 and Fig 2). *Ordinal* data provide ordering of the distinct values (i.e., a succeeding ordinal value implies greater quantity of the measurand) but provide no information about intervals between ordinal values (Eq 9 and Fig 2). We further classify continuous measurements which are both technically "quantitative" as quantitative or "semi-quantitative" given their relationship to a model measurand. *Quantitative* measurements are ordered values with defined intervals (i.e., the measured value determines, definitively, the quantity of the measurand). Researchers also designate measurements as *semi-quantitative* to convey that they possess less certainty than typical quantitative measurements. We define semi-quantitative measurements as consisting of an indirect proxy measurement that requires an uncertain (Eq 10 and Fig 2) or unknown (Eqs 11 and 12 and Fig 2) function $f_M$ to relate them to their measurand. The degree of uncertainty introduced by $f_M$ dictates how limited the semi-quantitative data is in mechanistic models of biology. Each data type reveals different insights about the cellular process and offer different contributions to the development of a mechanistic model.

In apoptosis signaling, nominal observations supported early research where it helped identify key components in the apoptosis signaling pathway [22]. Nominal apoptosis and survival outcomes–as indicated by morphology observations or DNA fragmentation data (Fig 2A)– helped determine two parallel signaling arcs that proceed following initiator caspase activation: mitochondria-dependent and -independent pathways[22]. These pathways trigger apoptosis by activating effector caspases [22]. We built an abridged Extrinsic Apoptosis Reaction Model (aEARM) [23], which represents these extrinsic apoptosis execution mechanisms as biomolecular interactions and where all parameters are identifiable (Fig 3A). Nominal measurements and observations guide mechanistic modeling by revealing salient structural elements of a cellular process but provide limited insight into the dynamics and complex regulatory cues of apoptosis signaling. Ordinal measurements have featured prominently in works investigating apoptosis signaling. They have uncovered clues about the dynamics and complex regulatory mechanisms of apoptosis. For instance, ordinal measurements of DISC (i.e., a ligand-dependent membrane bound 'death inducing signaling complex') components, initiator- and effector-caspases (Fig 2, *third row*), Bcl-2 family proteins (e.g., Bid), etc. revealed how cells resist

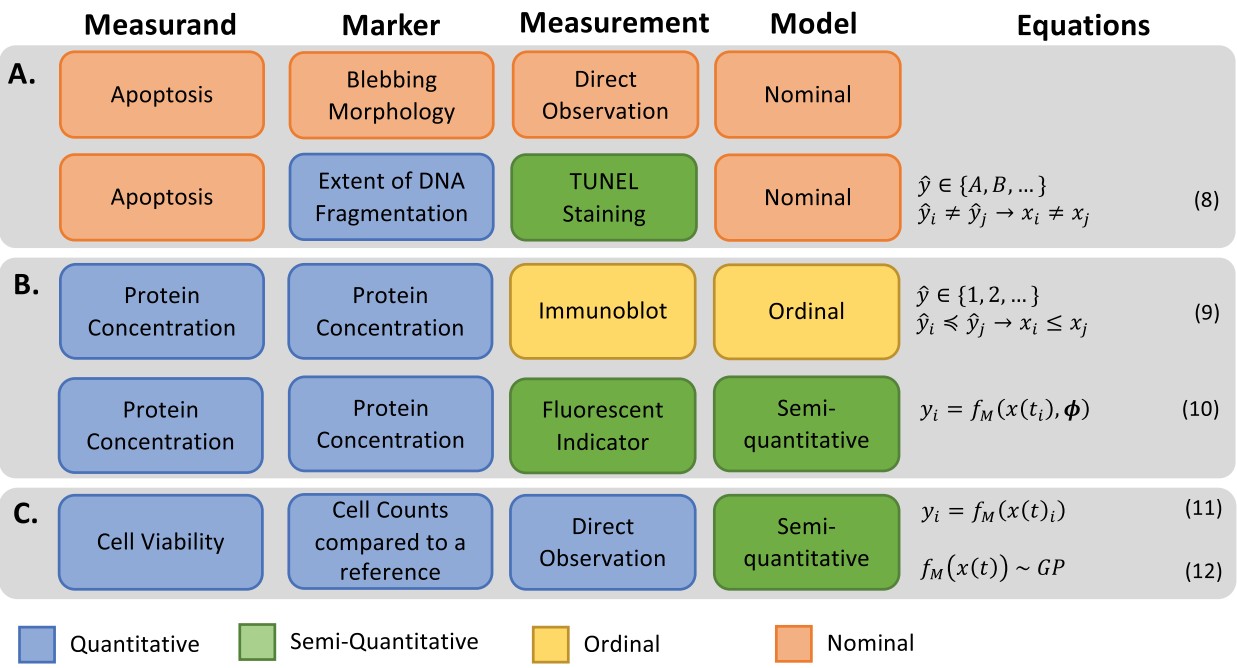

**Considerations for integrating data into mechanistic models using explicit measurement models:**
The measurement process consists of four components, each of which can impact how we integrate resulting data into mechanistic models. The measurand is the property we aim to quantify via an experimental measurement or assay. The measurand is associated (ideally tightly associated) with a marker, i.e., some component in the system that is directly accessible to the measurement assay. The measurement is the experimental assay that observes or measures the state of the marker (thereby conducting the measurement). Finally, the measurement has a model which defines how we interpret and apply the measured values to answer experimental questions. Implicit measurement models treat the measurement and corresponding measurand as identical (e.g., in the top row, nominal observations of apoptosis are conducted via direct observation of a nominal marker, blebbing morphology. In this case, the measurand and measurement are both nominal). Eq. 8 and 9 encode categorical constraints on the measurand, which we model as a a latent variable $(x_i)$ that can be encoded in a mechanistic model. We denote measurements as semi-quantitative if they lack a definitive relation to their measurand (e.g., $\hat{y} = x_i$). Eq. 10-12 describe measurement models that use a function $f_M$ to predict a value $y_i$ which can then be directly compared to observations $\hat{y}_i$ (e.g., via Eq. 6 in Box 1). In Eq. 11 $f_M$ maps the trace $x(t)$ from the $i$-th experiment to a predicted cell viability value $y_i$ for that experiment. Implicit measurement models encode $f_M$ as a simple equivalency relation, which works best for data that can be integrated into the mechanistic model without modification (typically quantitative data). More complicated measurement processes require explicit measurement models. How we define and apply measurement models depends on (A) the goals of the mechanistic model, (B) limitations of the measurement technology and (C) scope of the model. Here we present examples of each consideration using apoptosis measurement:

(A) **Goals of the mechanistic model dictates the measurement model:** The goal of aEARM, to predict and describe apoptotic signaling, motivates incorporation of apoptosis data. Apoptosis assays may use a quantitative marker (extent of DNA fragmentation) and measurement (TUNEL staining), but we would model this measurement as *nominal* apoptosis vs. survival because this fits the apoptosis related goals of the mechanistic model. If the goals of the mechanistic model focused on e.g., DNA fragmentation, this measurement would be modeled differently.

(B) **Limitations of the measurement technology dictates the measurement model:** Protein concentration is a quantitative variable. Technical challenges of biological complexity and heterogeneity however limit researchers to semi-quantitative or ordinal measurement of intracellular protein concentration. Biological and technical noise sources render immunoblot measurements ordinal (or ranked), so we model them as ordinal measurements. Similarly, fluorescent indicators of intracellular protein concentration are semi-quantitative, so we model them as such.

(C) **Scope encoded by the mechanistic model dictates the measurement model** Cell viability is a quantitative variable that can be measured quantitatively using available technologies. However, although cell viability measurements are relevant indicators of apoptotic signaling, they exist beyond the scope encoded by aEARM. We therefore model cell viability as an indirect proxy (i.e., semi-quantitative) measurement of apoptosis.

**Fig 2. Considerations for integrating different measurements and observations into mechanistic models.** Defining a measurement as (nominal (A), ordinal (B) or semi-quantitative (C)), and applying that measurement to a mechanistic model, requires consideration of the measurement *process*. Properties of the measurement such as the measurand (the property targeted by experimental measurement), its marker(s), the measurement technology and the goals of the experiment or model, dictate how we can apply the measurement within our mechanistic model. We discuss three scenarios (A., B., and C.) and their impact on how we model the resulting measurement.

apoptosis by limiting (but not eliminating) pro-apoptotic cues [24]; the sub-maximal pro-apoptotic signaling presents as delay in the dynamics of caspase activation [24]. To better understand caspase activation dynamics and its effect on apoptosis and survival, we need mathematical models of the apoptosis signaling dynamics. Ordinal measurements, however, do not readily support a mathematical description of apoptosis signaling dynamics. We note, here, that time-course measurements (e.g., ordinal measurements of the DISC at different times post-ligation) offer more toward an understanding of the dynamics of a biological process than time-independent (endpoint, steady-state, etc.) measurements. Ordinal and nominal measurements and observations can be time-dependent and -independent (depending on the experiment). Emerging work has leveraged ordinal and nominal measurements in the development of mathematical models of biological signaling but the weak constraints encoded by these measurements (Eqs 8 and 9 and Fig 2) add uncertainty and potential bias to the modeling process. This motivates more quantitative (or semi-quantitative) experimental measurements. We note, some semi-quantitative (e.g., fractional cell death) measurements lack a clear or definitive relationship to their measurand (e.g., the apoptotic susceptibility of the treated cells). While these measurements can encode tighter constraints on the measurand (Eqs 11 and 12 and Fig 2), the indirectness of the measurement provides little, or no, information about the function $f_M$ that could describe these constraints. These semi-quantitative measurements therefore occur in the early investigative stages and offer similar insights as nominal and ordinal measurements.

Technical challenges confine our quantitative and semi-quantitative measurements (required for traditional mechanistic model calibration methods) to just a few apoptotic signaling proteins. Fluorescence indicators of caspase activity (and by proxy, caspase substrate cleavage) enabled time course measurements of Bid and PARP cleavage dynamics (Fig 3B *top row*) [21]. They revealed pro-apoptotic activation of Bid and PARP, in TRAIL induced apoptotic HeLa cells, follows sigmoidal dynamics with delays and switch times that are sensitive to various regulatory factors. These measurements provide the details necessary for a mathematical description of apoptosis signaling dynamics and complexity. Our mathematical model aEARM captures the events from initial death ligand cue, initiator caspase activation, BID truncation (tBID), mitochondrial outer membrane permeabilization (MOMP) and eventual PARP cleavage (cPARP), as shown schematically on Fig 3A [23, 24–28]. The model was calibrated to the above fluorescence data, as described by Ortega et al [21, 23]. Semi-quantitative measurements like fluorescence, though better defined than less certain semi-quantitative measurements (e.g., fractional cell death), still lack a definitive estimation of the measurand because their interpretation requires mathematical manipulation, typically through a scaling function (Eq 10 and Fig 2), which can also add uncertainty and potential bias. Conversely, quantitative measurements have a definitive (e.g., equivalency) relationship with their measurand, and can be used directly in a model without further modifications thus minimizing the uncertainty and bias introduced to the model from this rarer measurement's interpretation. Therefore, the specific type of measurement and its interpretation can add significant uncertainty and potential bias to the mechanistic explanation of a given process.

To study the predictive power of mechanistic models supported by different types of measurements, we encode explicit measurement models that learn data-driven interpretations of

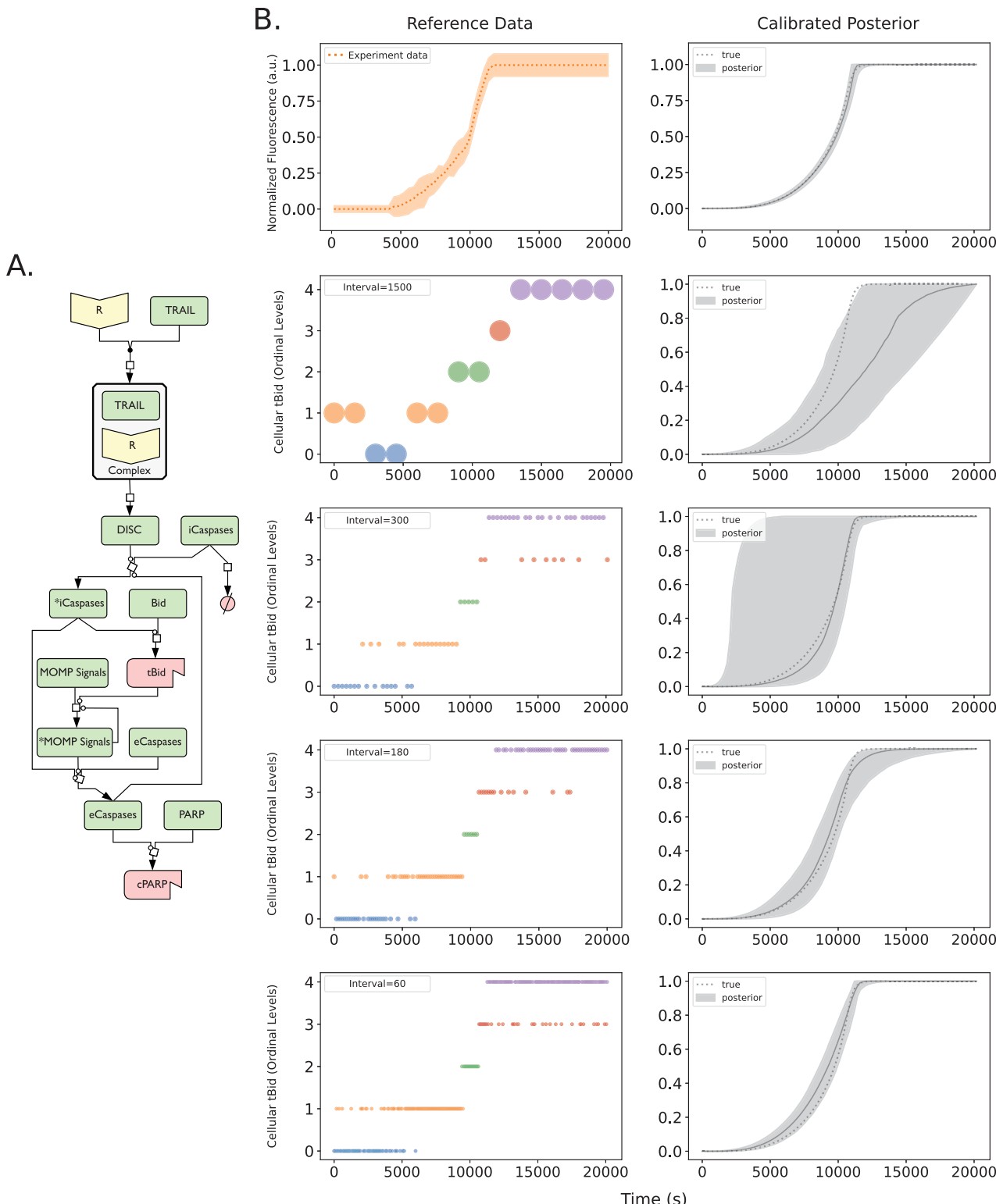

**Fig 3. Predicted Bid truncation dynamics of aEARM trained to different sized ordinal datasets.** Multiple Bayesian optimizations were run on the A.) abridged Extrinsic Apoptosis Reaction Model (aEARM) using different sized ordinal dataset to probe how dataset size influenced certainty of aEARM predictions. B.) Initiator caspase reporter (IC-RP) fluorescence time-course measurements (at 180s intervals) were measured (top left) as a proxy for truncated tBid (data from Albeck et al[21]). The plot shows the mean (dotted orange line) ± 1 standard deviation (shaded region) for each time point. The 95% credible region (top right) of posterior predictions (shaded region) for tBID concentration in aEARM, calibrated to fluorescence

measurements of IC-RP and EC-RP (See also Fig C in S1 Text). The median prediction (solid-line) and ground truth (dotted line) tBID concentration trajectories are shown. In the next four rows (from top to bottom), Ordinal measurements of tBID (left) at every 1500, 300, 180 and 60s interval, respectively. The 95% credible region of predictions (shaded region), median prediction (solid line) and true (dotted line) tBID dynamics for aEARM calibrated to ordinal measurements of tBID and cPARP occurring at every 1500, 300, 180 and 60s timepoint are plotted in plots on the right. The plots for cPARP ordinal measurements and predictions are found in Fig A in S1 Text.

categorical measurements and enable use of Bayesian model calibration methods. The explicit measurement is a function that maps variables from the mechanistic model $x$ to the values expressed in the data $y_{obs}$. This function is often assumed or implied, particularly for direct semi-quantitative data that can more readily be applied to the model calibration. However, the application of nominal and ordinal measurement types to mechanistic models is not straight-forward, because the weak constraints they impose on the values of their measurand (Eqs 8 and 9 and Fig 2) leaves room for interpretation. However, this interpretation (as we show in the following sections) can significantly bias model-derived insights. Consequently, modeling efforts have relied almost exclusively on quantitative and direct semi-quantitative data. By contrast, the much more abundant non-quantitative datatypes are often ignored or used inappropriately.

Early modeling efforts interpreted nonquantitative data as a series of arbitrary surrogate quantities for the ordinal or nominal values in a corresponding dataset [14]. More recently, discrete boundaries on the values of the measurand were imposed along with a distance metric to describe how well the mechanistic model satisfies nominal or ordinal constraints in the non-quantitative data [10–13]. These approaches reveal the potential value of nonquantitative data for mechanistic model calibration, but the often-*ad hoc* nature of these constraint-based measurement models has been an overlooked source of model bias. To avoid biases from the interpretation of non-quantitative datatypes, measurement models can assign free parameters to represent the unknown position of boundaries on the values of the measurand (or other properties of the categorical measurement) [10, 11]. Our measurement model takes this data-driven approach in that it possesses free parameters that are calibrated to match data; this lets us replace *a hoc* assumptions about the measurement with a data-driven parametrization, and thereby calibrate mechanistic models whose prediction better reflect the constraints imposed in the data.

The additional free parameters in data-driven measurements models, however, can be a source of variance that confounds the model calibration; this represents an inherent limitation in categorical data. We therefore, employ Bayesian methods and uncertainty estimation to examination how nonquantitative and quantitative measurements affect mechanistic model's *a posteriori* predictive power. To accomplish this, our measurement model replaces discrete boundary-based measurement models and distance metrics with predictions of the *probability* (Eq 4 and Fig 1) of the ordinal or nominal value given values of the measurand. This enables easy formulation of a likelihood function and application of Bayesian optimization methods that utilize MCMC sampling. In our approach, the measurement model is a mathematical construct that represents the measurand through machine learning probabilistic classification and regression models, whose free parameters are estimated simultaneously with the free parameters of the mechanistic model during calibration (Fig 1). Our measurement model effectively describes the probability of the categories encoded in the non-quantitative data given values of the measurand (Eq 4 and Fig 1). The measurand, in our case, is encoded in the mechanistic model. For example, the measurement model (Eq 4 and Fig 1) can use ordinal logistic classifiers to model the probability of a categorical value as a function of variable(s) encoded by the mechanistic model. We use a linear logistic model of the ordinal categories to minimize the number of added free parameters (i.e., to just $J$ for a dataset with $J$ categories). Similarly,

logistic classifiers can model the probability of cell death or survival observations given specific states of the mechanistic model. In the calibration process, the measurement model is an explicit intermediate step between simulation of the mechanistic model dynamics and calculation of the likelihood (Fig 1B). As described in the *Methods Section 2*, this approach uses the Python based PySB models-as-programs framework and PyDREAM, a Python implementation of the DREAM$_{(ZS)}$ algorithm to sample posterior values of models' free parameters. However, other model-building and parameter sampling (or optimization) algorithms could be employed by the user. In what follows, we examine the impact of different measurement modalities and interpretations on mechanistic model constraints in apoptosis execution. This work motivates an approach that could be generalized to any mathematical model to rigorously integrate quantitative and nonquantitative data types.

## Uncertainty of models calibrated to ordinal and nominal measurements of apoptosis dynamics

To date, molecular biology investigations of intracellular signaling processes and their mechanisms predominantly report nonquantitative measurements. However, it is unclear exactly how well these measurements support the development of mechanistic models. We therefore asked how various measurement datatypes impact the certainty and accuracy of model calibrations. Specifically, we explored how to adjust the size and make-up of nonquantitative datasets to better support mechanistic inferences. The aEARM was previously calibrated to fluorescence data by Ortega et al, from which we extracted the maximum *a posteriori* vector of aEARM kinetic rate parameters [23]. We centered the prior for these rate parameters on this maximum *a posteriori* parameter vector. We subsequently calibrated the parameters to the same fluorescence data referenced in Ortega et al [21,23]; the resulting posterior predictive region for tBID dynamics of aEARM is shown in Fig 3 (B, top row). As expected, the data can effectively constrain the model and the 95% credible region of posterior predictions for tBid dynamics falls within the data uncertainty region. We also used the maximum *a posteriori* parameter vector as a baseline (reference) to generate ordinal datasets for tBID and other aEARM variables as described in *Methods Section 3* and shown in Fig 3B (bottom four rows). These synthetic datasets could be considered as numerical representations of a time-course western blot dataset. We then calibrated aEARM kinetic rate and measurement model parameters to the ordinal and nominal datasets.

As shown in Fig 3B, ordinal datasets accurately predicted quantitative predictions of "ground truth" dynamics for tBID. The 95% credible region of posterior predictions of tBID dynamics of aEARM trained to these ordinal datasets each contained "ground truth" dynamics for tBID. We also use the area bounded by the 95% credible region of posterior predictions of tBID as a measure of model certainty; with a smaller area indicating higher certainty. The ordinal dataset containing measurements at every 25-minute interval (i.e., typical of time-dependent western blot datasets), however, did not significantly constrain the posterior predictive regions of these dynamics (Fig 3B second row). Increasing the number of measurements, however, increases the certainty of the posterior predictions of tBID dynamics; this certainty approaches that of the typical semi-quantitative (fluorescence) dataset that has an area of 2.7, when then the number of ordinal measurements is increased threefold, which had an area of 6.2. The areas bounded by the 95% credible region for each ordinal time-course dataset is described in the Fig 3B (Bottom two rows).

To explore the impact of nominal data on model optimization, we again used the maximum *a posteriori* parameter vector from the fluorescence optimized aEARM as a baseline (reference) to generate nominal datasets akin to an apoptosis execution observation as described in

*Methods Section 3*. Previous work has described how features of apoptosis signaling dynamics can predict cell death vs survival [20]. The generated nominal dataset describes binary cell-fate outcomes that emerge because of extrinsic apoptosis signaling dynamics. We encode this information in a nominal measurement model as described in *Methods Section 2.3*. Parameters of aEARM and the free parameters encoded in the measurement model were jointly calibrated to a synthetically generated dataset of 400 survival vs death outcomes as shown in Fig 4A (left). As shown in Fig 4A (right), the binary cell-fate data minimally constrain the posterior predictive region of tBID dynamics relative to the prior constraints on the model. This is expected as the binary cell-fate datatype essentially condenses complex apoptotic signaling dynamics to a single categorical value. Fig R in S1 Text shows the 95% credible region of posterior predictions of normalized tBID dynamics of aEARM trained to published fractional cell death data. These data differ from synthetic ordinal and nominal datasets in that they lack a known reference to "ground truth" dynamics and should therefore not be compared to the plots in Figs 2 and 3.

In lieu of its limited ability to constrain mechanistic models, modeling efforts understandably disregard nominal data. However, we hypothesized that combining nonquantitative datatypes and covering multiple variables in the model could improve model certainty. To explore the effect of multiple data type combinations on model calibration, we again optimized the aEARM model parameters, but this time to a dataset containing nominal and ordinal measurements. As described in *Methods Section 3*, we added a synthetic dataset containing 61 ordinal time-course measurements for the DISC complex to the nominal dataset described above (Fig 4B (left)). We modeled the likelihood of this combined dataset as the product of the likelihoods of the individual constituent datasets (see *Methods Section 2* for details). In Fig 4A and 4B (right), we see the nominal and ordinal datasets yields larger 95% credible regions for the posterior predictions of tBID dynamics. However, (in Fig 4C) the combined dataset better constrained the posterior predictions of normalized tBID dynamics than either dataset alone, with a 95% credible region area of 26.5 (compared to 55.0 and 56.4 for the ordinal and nominal datasets alone). Therefore, the model uncertainty stemming from only using tBID nominal data was decreased by including more detailed upstream measurements. However, the contribution of DISC ordinal data alone was comparable to that of the tBID nominal data in isolation (Fig 4B (right)). This data suggests that distributed measurements across multiple variables in a pathway yield synergistic effects on calibrated model accuracy and certainty. Though these findings are specific to our aEARM model, they provide heuristic insights to consider in other mechanistic models.

## Data-driven measurement model as an indicator of model bias

Traditionally, applying quantitative or semi-quantitative data to a mechanistic model has been relatively straightforward as they typically follow a well-established and simple relationship between the measurement and the measurand. However, for non-quantitative data, measurement uncertainty can prompt researchers to make assumptions about the relationship between measurement and measurand, which may negatively impact in the resulting mechanistic model. We therefore asked how the encoding of assumptions into our models of non-quantitative measurements could impact mechanistic model calibrations. To attain this goal, we calibrated aEARM kinetic rate parameters to the ordinal dataset, but this time we replaced the flexibility of free parameters in our measurement model with fixed *a priori* parameterizations (i.e., preset values not calibrated to data). We also leveraged the Bayesian feature of our approach and encoded our assumptions as priors on the measurement model's free parameters. In this way, we more flexibly encode our prior knowledge and assumptions about the

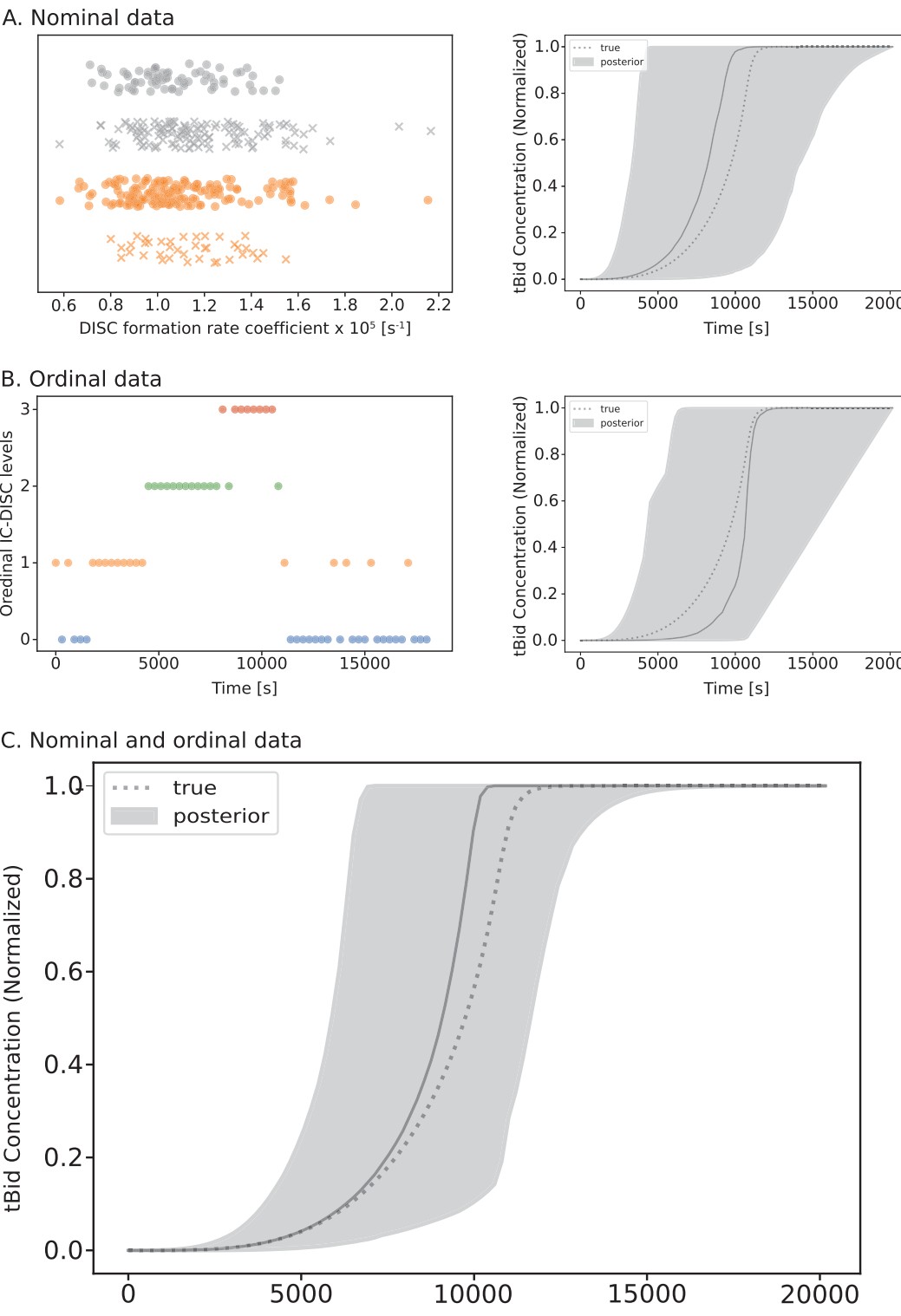

**Fig 4. Predicted Bid truncation dynamics of aEARM trained to nominal and ordinal datasets.** A.) Nominal cell death (x) vs survival (o) outcomes data for cells treated with 10ng/mL (orange) and 50ng/mL (grey) of TRAIL and with known relative values of DISC formation (x-axis). The 95% credible region (shaded region) of posterior predictions of tBID dynamics of aEARM calibrated to nominal data (right plot). The median prediction (solid-line) and true (dotted line) are also plotted. B.) Ordinal measurements for initiator caspase-DISC colocalization (IC-DISC) at 300s intervals (left plot). The 95% credible

region (shaded region) of posterior predictions of tBID dynamics of aEARM calibrated to ordinal IC-DISC data (right plot), and C.) of aEARM calibrated to nominal *and* ordinal IC-DISC data. The median prediction (solid-line) and true (dotted line) were also plotted. The fit to IC-DISC data is shown in Fig G S1 Text.

measurement into the measurement model. We tested four situations: (*i*) fixed parameters, a case where the measurement model is pre-parameterized by the user, presumably reflecting full confidence in their assumptions about the measurement; (*ii*) strong prior knowledge, a case where there is strong belief in the assumed values of the measurement model parameters; (*iii*) weak prior knowledge, a case where there is only weak belief in the assumed values of the measurement model parameters; and (*iv*) no prior knowledge, that is no constraints on the measurement model parameters.

Fig 5A and 5B show the ordinal class probabilities for tBID as modeled by (i) two distinct pre-parameterized measurement models. In case 1, lowest and highest categories correspond to a narrow range of tBID values, while the three internal categories each account for roughly 1/3$^{rd}$ of the tBID range. This parameterization might aim to account for effects of sensitivity and saturation on the measurement. In case 2, all five ordinal categories each account for 1/5$^{th}$ of the range of tBID values. The right panels in Fig 5A and 5B show the assumed relationship between tBID concentration and probability of each ordinal category. Fig 5A and 5B (left plots) also show posterior predictions of tBID dynamics by aEARM calibrated to the ordinal dataset using these fixed pre-parameterized measurement models. The different measurement model pre-parameterization produced markedly different posterior predictions of tBID dynamics by the resulting aEARM calibrations. This raises potential concerns that assumptions in our interpretation of the measurement can artificially influence our interpretation of the mechanism.

In Fig 5B, the 50% probability boundary between adjacent categories occurs at every 0.2 interval; dividing the [0,1] range of tBID values into five equally spaced ordinal categories. Shown in Fig 5C–5E, we represent this as a *flexible* assumption by encoding it in our priors (ii–iv) i.e., Cauchy distributions centered at every 0.2 interval or a uniform distribution (as detailed in *Methods Section 5*). The smaller the scale–more narrowly focused–prior distributions reflect less flexibility in the free-parameter and a stronger belief in our prior assumptions. Fig 5C–5E shows the posterior predictions of tBID dynamics of aEARM calibrated to the ordinal dataset using increasingly more constrained priors on the measurement model parameters. The resulting posterior predictions of tBID dynamics were all less constrained than that of aEARM calibrated using fixed pre-parameterized measurement models (Fig 5A and 5B) but, they were more accurate as they contained the "ground-truth" tBID dynamics. Strongly constrained priors on the measurement model parameters (ii) produced a less certain mechanistic model; as indicated by its wider 95% credible region of posterior predictions of tBID dynamics (Fig 5E). The posterior distributions of measurement model parameters were spread out enough to give significant support of both the "ground truth" and the *a priori* assumed parameter values. This uncertainty in the measurement model parameter distributions translated into a less certain measurement model and less certain predictions of tBID dynamics. Weaker constraints on the measurement model parameters were encoded via larger scale prior distribution (Fig 5D). In Fig 5D we see these prior distributions, while centered on our *a priori* assumptions, includes the "ground truth" parameters. The posterior distributions of the measurement model parameters were therefore more constrained; likewise, the measurement model and posterior predictions of tBID dynamics had more certainty. This is also observed, in Fig 5C, the case where no prior assumptions were applied (as modeled using uniform priors on the measurement model parameters) to the measurement model parameters (iv). The

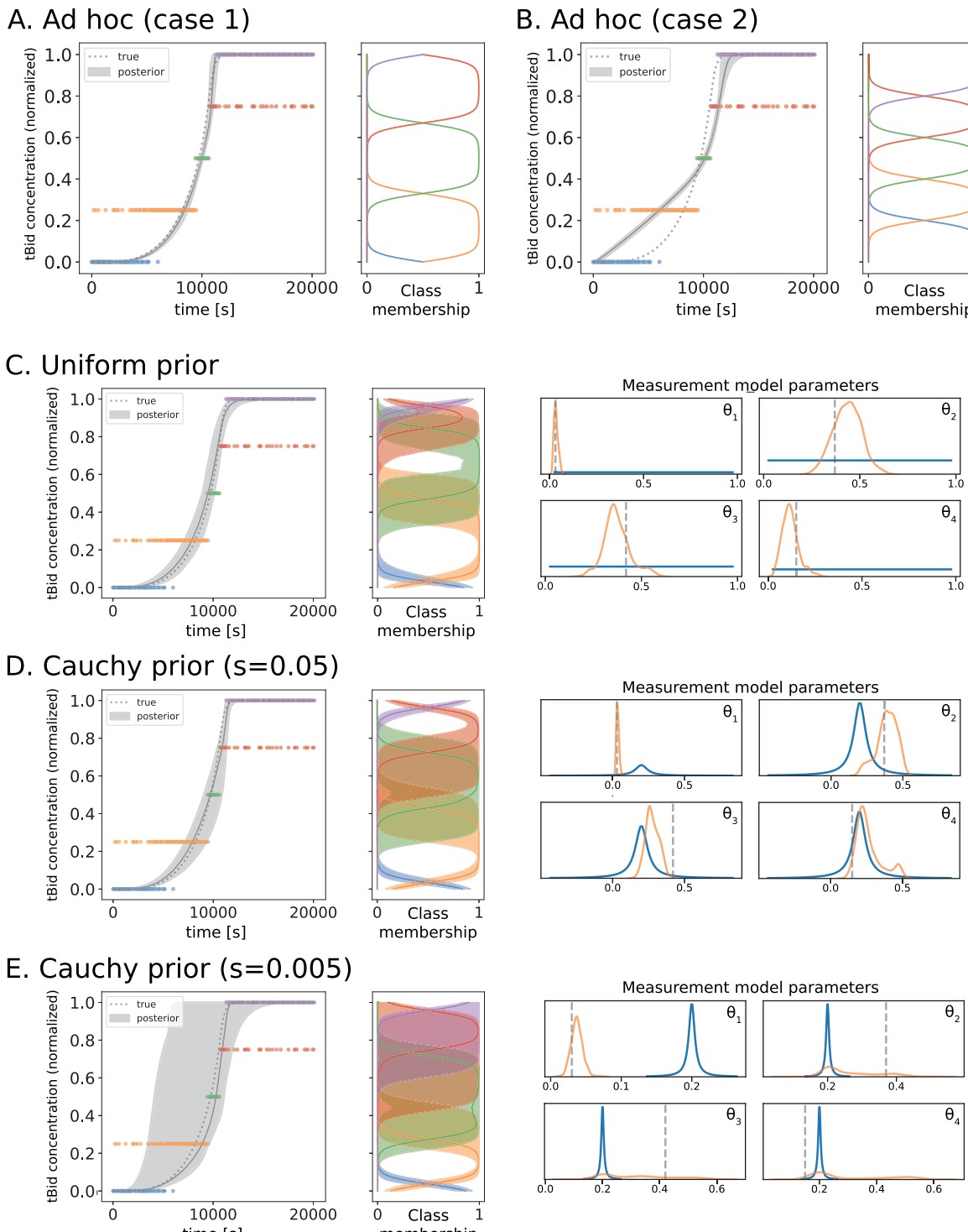

**Fig 5. Predicted Bid truncation dynamics of aEARM trained to ordinal data using different measurement model parameterizations.**
A.) and B.) The 95% credible region of posterior predictions (shaded region) of tBID dynamics for aEARM calibrated to ordinal measurements two fixed parameterizations for the measurement model (see Table C in S1 Text). The adjacent panels plot the measurement models predicted probability of class membership (x-axis) as a function of normalized tBID concentration (y-axis). C.) D.) and E.) The 95% credible region of posterior predictions (shaded region) of tBID dynamics of aEARM calibrated to ordinal measurements uniform, Cauchy (scale = 0.05) and Cauchy (scale = 0.005) prior distributions for the parameterizations of $\theta_j$ (the distance between offset $\beta_j$ and the preceding offset $\beta_{j-1}$) for the measurement model, respectively. In each, the median prediction (solid line) and true (dotted line) tBID dynamics are also shown. The adjacent panels give the 95% credible region of posterior predictions of the probability of class

membership (x-axis) as a function of normalized tBID concentration (y-axis). The left and their adjacent panels share the y-axis (normalized tBID concentration) Four accompanying plots show the prior (blue), posterior (orange) and true (dashed line) values of measurement model parameters.

accuracy of the predictions of tBID dynamics comes from the flexibility of the data-driven measurement models' parameters. This flexibility enables optimization (or prediction) of key properties of the measurement given the data. Fig 5C–5E (right panels) shows posterior predictions of the probability of ordinal class membership (modeled as a function of predicted cellular tBID content); these predictions are accurate in that they contain "ground truth" probabilities. Using this approach, we calibrated more accurate models of mechanism by simultaneously learning a more accurate model of the measurement. This motivates us to further explore the data-driven measurement model as a potential new avenue for insights.

## Mechanistic insights from data-driven measurement models

We have shown thus far how a machine learning measurement model can reduce uncertainty and increase accuracy in model calibration. Through mechanistic model calibration to categorical data, we effectively employ machine learning classifiers to constrain mechanistic model dynamics to a corresponding categorical phenotype. We leverage the learnable aspect of our measurement model and model calibration method to employ the measurement model in reverse: to learn how properties of a biological mechanism predict, drive, and define a particular phenotype. This kind of knowledge would be essential for model-driven experimental data acquisition and model-guided validation.

To demonstrate this concept, we calibrated aEARM to synthetic nominal cell survival vs death data using a measurement model that estimated the contribution of variables in aEARM to the cell survival vs. death predictions. The survival vs death dataset was synthesized based on maximum log-rate of change of tBID and the time at which the rate of change maximized; these features were encoded into the measurement model, but their contribution was represented as a free parameter. In addition, the measurement model also considered the potential contribution of an unrelated variable (i.e., concentration of a reactant in reactions that occurred independently of the cell death ligand). Jointly calibrating aEARM and this measurement model to cell survival vs death data allowed data-driven predictions of how variables encoded in aEARM relate to cell survival vs death. Fig 6 shows posterior predictions of the values of potential predictors of cell survival vs death. The shaded region marks the 95% credible interval for the line marking 50% cell survival probability. Fig 6 (bottom row) provides the posterior distribution of weight coefficients for each the features encoded in the measurement model. (Larger absolute values of the weight coefficient indicate greater importance of the feature.) The calibrated measurement model correctly identified time at maximum Bid truncation as the most important predictor of our synthetic cell survival vs death data; and the unrelated variable as the least important predictor. Calibration of aEARM to the mixed dataset, described in the previous section, yielded a measurement model that equivalently predicted identified time at maximum Bid truncation as the most important predictor of cell survival; and the unrelated variable as the least important predictor. Calibration of a mechanistic model to categorical phenotype data, using data-driven measurement models, enabled correct identification of predictors (and potentially drivers or markers) of categorical phenotypes. The data-driven probabilistic measurement model we propose in this research was essential to this finding.

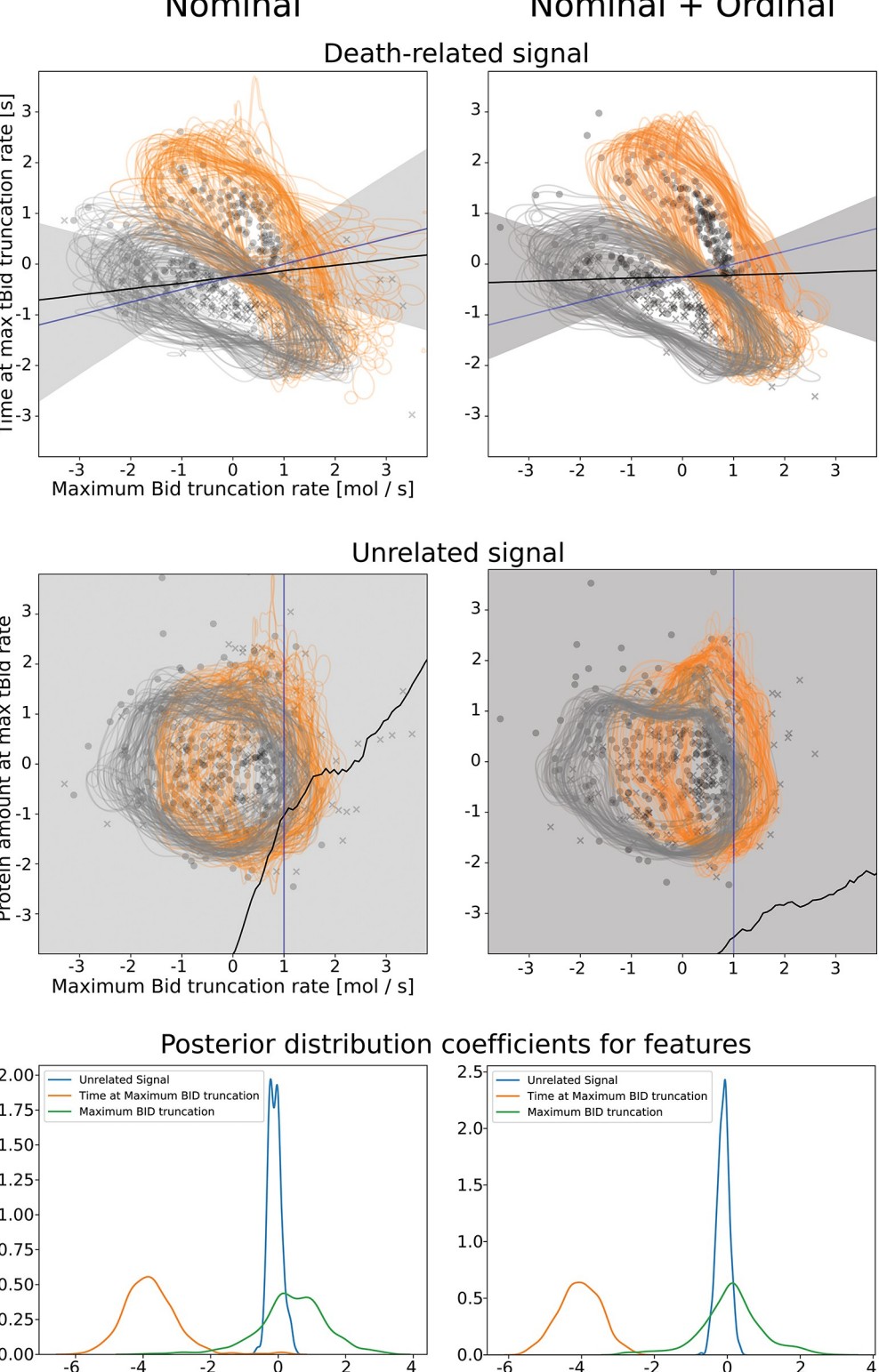

**Fig 6. Measurement model predicts features of cell death vs. survival using aEARM calibrated to cell death datasets.** Normalized predicted values of the features used in the cell death vs. survival measurement model–the x-axis is the maximum Bid truncation rate, and the y-axis is the time at maximum Bid truncation rate (top row) or an unrelated non-apoptotic signal (middle row)–for corresponding to observed cell death (x) and survival (o) outcomes.

These feature values are modeled by aEARM parameterized by 100 parameter vectors randomly drawn from the posterior; for each parameterization, 5 out of the total simulated population of 400 cells were plotted. The grey and orange curves, in these plots, are 0.05 contours for the estimated density of simulated cell populations produced for each of the 100 parameter vectors–grey and orange correspond to 50 and 10ng/ml TRAIL treatments, respectively. The measurement model predicts a probability of cell death vs survival based on simulated values of the above features. The lower right region of the plots in the top row. (i.e., early maximization of Bid truncation and higher maximal Bid truncation rates) is associated with higher probability of cell death. The shaded region is the 95% credible region of the posterior prediction of the line marking 50% probability of cell death or survival. The black and blue lines are the median predicted and true 50% probability lines, respectively. The bottom row plots the posterior distributions of the weight for each feature (i.e., the product of the slope term and feature coefficient encoded in the measurement model): maximum Bid truncation rate (green), time at maximum Bid truncation (orange) and unrelated non-apoptotic signal (blue). Plots in the left column are predictions of aEARM calibrated to the cell death vs. survival dataset. Plots right column were those of aEARM calibrated to the cell death vs survival + ordinal IC-DISC combined dataset.

## Application of data-driven measurement model to published fractional cell death data

To explore a practical application of the data-driven measurement model paradigm, we calibrated the aEARM parameters to published fractional cell death measurements (Fig 7A) by Wajant et al [29]. These measurements were taken in HeLa cell in which apoptosis was inhibited through graded expression of a dominant negative FADD. Wildtype, low- and high-dominant negative FADD genotypes were modeled (see *Methods* *Sections 2.4 and 4*) via free parameters, $\delta_{low}$ and $\delta_{high}$, which represent diminished rates of DISC formation relative to wildtype $kc_0$ in aEARM. These free parameters were calibrated along with the parameters of aEARM (and supporting the measurement model). Fig S in S1 Text shows the posterior distribution of these parameters.

Fractional cell death measurements are indirect semi-quantitative measurements since they lack a clear mapping between the measured values and underlying cell death signaling dynamics. Specifically, there exists limited information about which features of the cell death signaling *dynamics* predict fractional cell death, and no established function to map those features into fractional cell death values. We therefore used Gaussian process (GP) regression to model the fractional cell death measurements. GP regression does not require we commit an assumed functional form for our measurement. Instead, it learns the posterior distribution over all functions that fit the data. This lets us flexibly apply Bayesian data-driven methods to our model of fractional cell death. In this way, GP model is a data-driven probabilistic measurement model like those demonstrated in previous sections. Our GP measurement model $y(x) \sim GP(m(x), \kappa(x, x'))$ maps features of aEARM modeled tBID dynamics $x$ onto fractional cell death values $y(x)$. We used a radial-basis kernel (Eq 20), for $\kappa(x, x')$), with free parameters for the coefficient ($\sigma_f^2$) and variance matrix ($M$) and calibrated those parameters along with the rate parameters of aEARM. The variance matrix in this kernel ($M$) is the diagonal $\beta^{-2}$, where $\beta$ is the weight coefficient for the features extracted from tBID dynamics predicted by aEARM. The dynamics of tBID (as modeled by aEARM and others) are smooth, differentiable trajectories that can be accurately summarized via a small number of critical points (maximum, minimum, point of inflection, etc.). This low-dimensional critical points representation of tBID dynamics includes those features explored in Roux et al [20]. A single GP model was used for all data in the training set (see *Methods* Section 2.4 for more information). The resulting likelihood function (i.e., the GP marginal likelihood Eq 22) was used, with PyDREAM, to calibrate aEARM to fractional cell death measurements.

The posterior predictions of fractional cell death of the calibrated aEARM (Fig 7B) aligns with the training data (Fig 7A). Further, the calibrated model also accurately predicts fractional cell death observations outside of the training set: fractional cell death data in TAT-BID mutant HeLa cells (Fig 7C) as observed in Fig 7G. of Orzechowska et al [30]. These

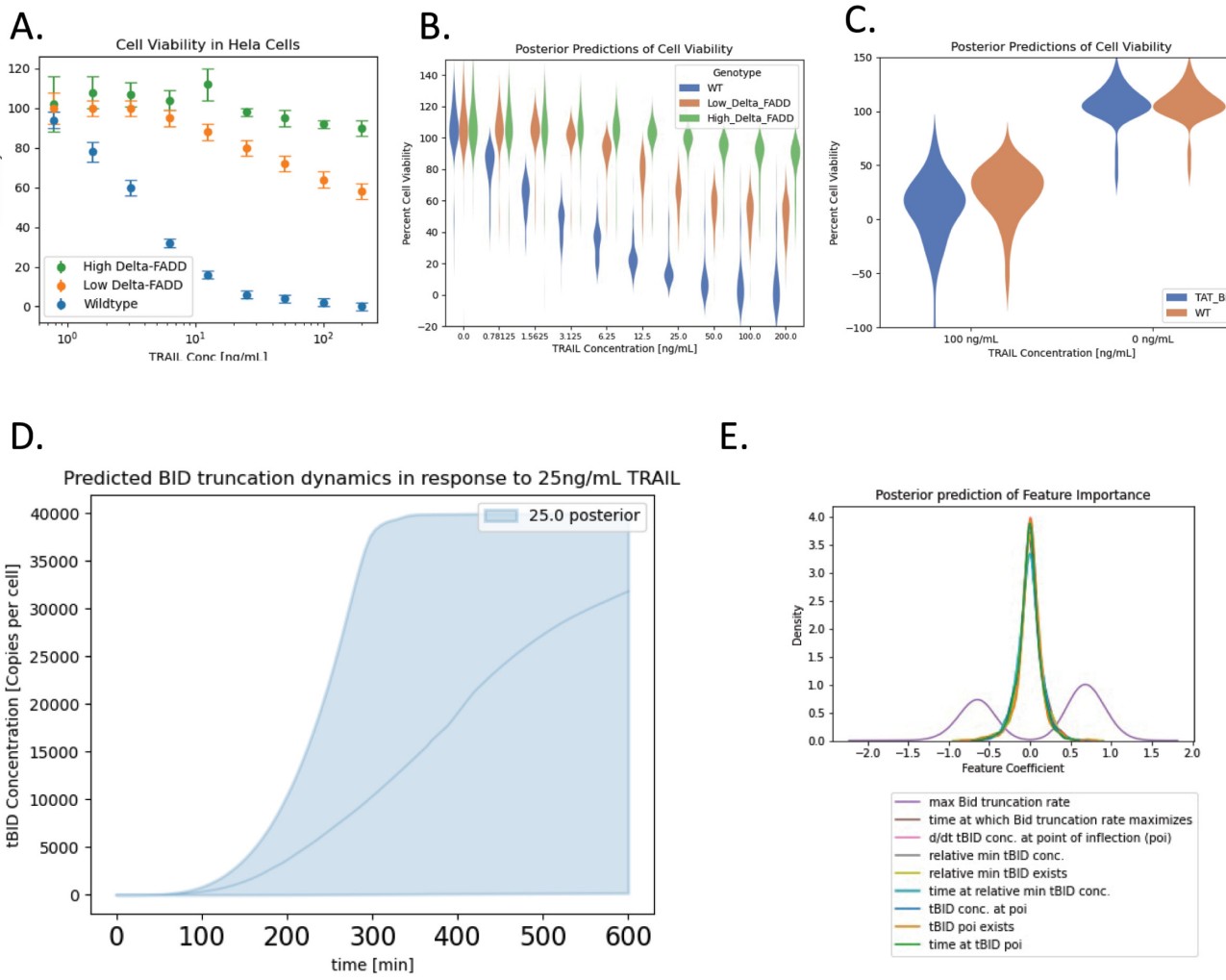

**Fig 7. Posterior predictions of aEARM trained to published fractional cell death data.** A. Fractional cell death in WT (blue) and high- and low- expression of dominant negative FADD (green and orange respectively) in HeLa cells treated with 0 to 200ng/mL TRAIL. These data come from Wajant et al. 1998 [29]. The aEARM and accompanying measurement model were calibrated to these data. B. The posterior predictions of the Gaussian process modeled mean fractional cell death values for WT and high- and low- expression of dominant negative FADD in HeLa cell treated with 0 to 200ng/mL TRAIL. C. Posterior predictions of the Gaussian process modeled mean fractional cell death values for WT and BID overexpressed (TAT-Bid) HeLa cells treated with and without 100ng/mL TRAIL. Fractional cell death predictions for these experimental conditions, which were excluded from our training dataset, correspond to fractional cell death measurements by Orzechowska et al [30]. The 95% credible region of the posterior prediction (D.) of tBID dynamics in cells treated with 25g/mL TRAIL. (E.) Posterior distributions of the weight for each feature extracted from tBID dynamics.

predictions, however, had much less certainty which might reflect limited information provided in fractional cell death data. The 95% credible region of the posterior predictions of tBID dynamics (in response to treatment with 25 ng/mL TRAIL) of the aEARM trained to published fractional cell death measurements (Fig 7D) similarly occupies a large (or uncertain) range of values. However, this range is comparable to the range of trajectories of a caspase activity fluorescent indicator in HeLa cells treated with 25 ng/mL TRAIL, observed in Roux et al (Reprinted in Fig S in S1 Text) [20]. For instance, the predicted maximal tBID dynamics exhibit points of inflection between 200 and 300 minutes, which accurately reflects the observed maximal tBID trajectories (in HeLa cells treated with 25ng/mL TRAIL) by Roux et al [20].

The fractional cell death measurement model considered critical point extracted from aEARM dynamics as predictive features of fractional cell death. Fig 7E. shows the posterior distributions of feature coefficient weight $\beta$. The features with greater displacement from zero indicate increased predictive value. The fractional cell death measurement model therefore identified the maximum Bid truncation rate as the most important predictor of fractional cell death. This is consistent with observations by Roux et al. that changes in the maximum fluorescence indicated caspase activity (proxy for BID truncation rate) imply sharp changes in observed probability of cell death (Reprinted in Fig S in S1 Text) [20]. This result underscores the capacity of the data-driven measurement model to reveal insights into the potential predictors and drivers of cellular phenotype.

## Discussion

We used data-driven measurement models to calibrate, using Bayesian methods, a dynamical model of biological mechanism to quantitative and nonquantitative data. Our approach allowed us to estimate posterior predictive regions for the calibrated models and to observe how the size of a dataset, its different measurement types, and our assumptions about the measurements affect model accuracy and certainty. Our findings support results from previous studies that suggest nonquantitative data are valuable for mechanistic modeling efforts [10–13]. For instance, a sufficiently large ordinal dataset can constrain the posterior predictions of a mechanistic model as much as quantitative dataset. However, for our model of apoptosis, far more nonquantitative data than is typically generated would be necessary for nonquantitative assays to match the information content of quantitative assays. In Fig 3B (second row), fourteen ordinal measurements of tBID–typical of common immunoblot measurements of intracellular biology–did not constrain the model around an accurate prediction of tBID dynamics. Instead, it took 24x as many ordinal measurements of tBID (336 measurements) to constrain the mechanistic model of apoptosis as well as the fluorescence dataset (112 measurements). We also found that datasets that combined categorical measurements of multiple variables in aEARM out-perform the datasets with measurements of an individual variable. These findings suggest one could overcome challenges posed by a dearth of quantitative data by devising experiments that, while nonquantitative, produce a larger number of diverse measurements that can cover multiple variables.

We also found the posterior predictions of our mechanistic model were sensitive to we encode assumptions in the measurement model, about the relationship between measurement and measurand. All measurements possess uncertain (or unknown) properties, but this uncertainty has a pronounced presence in nonquantitative measurements. The limitations of nonquantitative data exist because they impose less informative constraints on models, and this leaves room for potentially biasing assumptions and/or uncertainty. Uncertainty in nonquantitative measurements drives the, often unacknowledged and implicit, assumptions about the relationship between measurement and measurand (i.e., between data and model). With the proposed Bayesian calibration framework, we could observe how assumptions about measurement affected the uncertainty and accuracy of the posterior predictions, in essence providing a measurable quality of how well the model can make mechanistic predictions. We found that inaccurate *ad hoc* assumptions about the measurement could produce models that suggested, with a higher degree certainty, an inaccurate prediction (Fig 5B). This finding suggests that *ad hoc* assumptions about measurements can lull practitioners into a false sense of confidence about the model and the data. This concern also motivated Schmiester and co-workers to avoid certain *ad hoc* assumption by assigning free parameters to the constraints imposed in their model calibration approach [11]. However, without the use of Bayesian model calibration

methods, modelers might miss the impact of the added free parameters on the uncertainty of mechanistic models trained to these nonquantitative datatypes. Further, Bayesian model calibration methods enable parameter sensitivity analyses that could facilitate coarse graining of (or removal of less impactful parameters from) the model [31].

Having a measurement model whose attributes are determined by data creates an opportunity to *learn* new details about the relationship between a measurement and its measurand(s). For instance, could a model of biological mechanism plus cell phenotype observations data enable identification of cell phenotype predictors? To explore this, we encoded a small number of suspected cell-fate predictors into our measurement model and let the data (and the mechanistic model) determine, through model calibration, their respective contribution to phenotype. In doing so, model calibration using our data-driven measurement model performed feature selection to correctly identify the most important predictor or driver of cell death. We extend the data-driven measurement model to a practical example wherein published fractional cell death data were used to calibrate our mechanistic model of cell death. Here, we find calibration using these data produces a model that accurately predicts underlying cell death dynamics and correctly identifies features of those dynamics which best predict or drive cell death. The high uncertainty of the predictions of models trained to our cell death vs. survival data, however, reflects the limited information contained in nominal cell fate observations and fractional cell death measurements. Leveraging larger datasets (of multiple datatypes) would likely improve the certainty of the mechanistic model and measurement model predictions. The measurements models deliberately use simple supervised machine learning models that capture the salient features of the measurement while maintaining tractability of the model calibrations. These characteristics lets us map cellular dynamics to diverse cellular phenotypes without requiring prohibitively large datasets (and computing resources). The measurement model can readily extend to mappings from intracellular dynamics to extracellular processes (e.g., tumor growth). We restricted our modeling to published observations of TRAIL induced apoptosis in HeLa cells where experimental treatments produced a definitive impact on the proteins modeled in aEARM. However, models of other cell types of experimental systems etc. may be supported by a larger set of available data sources. In general, this kind of measurement model, which relates mechanism to cellular phenotype, can be used to predict phenotype outcomes and identify potentially informative experimental conditions from *in silico* perturbation experiments.

## Conclusions

This work explores how different nonquantitative measurement effect calibration of mechanistic models of biological processes. Essential to our analysis are measurement models that support Bayesian model calibration methods, flexibly encode knowledge and assumptions about the measurement, and learn new insights through their application. We show 1. More categorical data are required to constrain a mechanistic model. 2. Prior assumptions about the measurement can bias the mechanistic model predictions. 3. Measurement models can be devised to learn insights that potentially guide subsequent model-driven experimentation. Our measurement model is a proof-of-concept that can be improved upon in future work. We chose linear logistic classifiers, as they enable easy formulation of a likelihood function and application of Bayesian calibration methods, while minimizing introduction of additional free parameters. We note that *any* functional relationship between the mechanistic model and accompanying measurements (e.g., a universal function approximator) can be employed and tailored to the goals of the modeling effort. We constrained our measurement representation to small number of potential features to avoid complications of high dimensionality to our

machine learned measurement model. However, dimensionality reduction and feature learning (e.g., PCA) can, in theory, be integrated into the measurement model's preprocessing and/or model calibration workflow. Possibilities for integrating more complex machine learning into models of measurement will depend on dataset size, computational power, and modeling goals.

Our work expands the concept of measurement models to the mechanistic modeling paradigm. Measurement models have their origin in social sciences and statistics [32]. They also appear in more quantitative applications; some recent examples include management, manufacturing, and computer vision [33–35]. These measurement models can take on more complexity than the examples we provided, depending on the unique needs of the problems in these areas. The use of measurement models in these areas is motivated by a desire to define and quantify observations of nuanced and/or subjective phenomena; and connect those observations to an underlying theory. Our use of the measurement model to learn phenomenological links between an underlying process and qualitative biological observations is inspired by the use of measurement models in these fields [32–35]. Future, demonstrations of this concept might employ causal formalisms to distinguish between molecular markers and molecular drivers of a phenotype. Biology, being "harder" than social sciences, but arguably "softer" than physics will straddle the technical domains of both. As a field, we face the same challenge as these social sciences given that our mechanistic models are situated within a larger context of explaining nuanced and subjective biological phenomena (e.g., cell-fate, morphology, physiology and overall health vs. pathology). As practitioners, we never encode *everything* into our mechanistic models; instead, there is always some aspect of the model (or its interpretation) that aims to connect back to these relevant biological phenomena. This fact ultimately motivates our application of data-driven probabilistic measurement models in our mechanistic models of intracellular biology.

## Methods

### 1. Extrinsic apoptosis reaction model

We built an abridged extrinsic apoptosis reaction model (aEARM) and trained it using PyDREAM to normalized fluorescence time-course data, as describe by Ortega et al. [23]. We built this abridged version of EARM to simplify convergence of Bayesian calibration algorithms and thus make feasible probability-based predictions on the model-data relationship [23]. The aEARM abstracts detailed mitochondrial reactions from the original model as two sequential mitochondrial outer membrane pore (MOMP) "signal" activation steps. In addition, apoptosome formation and effector caspase activation reactions take place in a single activation step. The aEARM does capture key dynamic characteristics, such as the snap-action delay dynamics of apoptotic effector molecules that is observed empirically [21]. For this work, three additional non-apoptotic species were encoded and linked via feedback activation and inactivation loops to test whether our data-driven measurement model could discriminate between drivers and non-drivers of apoptosis. (S1 Data). These additional species and reactions do not interact with any species or reaction in the aEARM model. The aEARM was encoded using rule-based modeling python package PySB [36].

The aEARM parameters–initial conditions and rate coefficients–were adapted from the previously developed EARM and/or calibrated to fit available fluorescence data. Initial conditions parameters were lifted from the previously developed EARM (Table A in S1 Text). Previous work characterized extrinsic heterogeneity in the expression of proteins and its effect on apoptosis. To model extrinsic heterogeneity in apoptosis signaling, initial values of certain species (Table A in S1 Text) were sampled from a log-normal distribution such that its mean

equaled that in Table A in S1 Text and coefficient of variation was 0.20. Rate coefficients were calibrated (described in *Methods Section 5*) to fit normalized fluorescence time-course measurements of initiator and effector caspase reporter proteins (IC-RP and EC-RP respectively).

**1.2. Integrating aEARM dynamics.** Snap-action delay dynamics present challenges for Ordinary Differential Equation (ODE)-based models, as they feature rapid non-stiff to stiff transitions during integration. For this work we employed the LSODA integrator (from scipy, via the PySB solver suite), suitable for non-stiff/stiff systems [37]. However, we found that particularly poorly behaved parameter vectors could prolong integration evaluations in LSODA. Integrator settings were adjusted for efficiency and accuracy of integration as follows: mxstep (2^20), atol (1e-6 default), rtol (1e-3 default). The aEARM was integrated over a linear space of 100 time-points spanning 0 to 20160 seconds, in direct correspondence with the fluorescence time-course data [21]. Additional time-points in the data were obtained via linear interpolation.

## 2. Measurement models and likelihood functions

Likelihood formulations incorporated a measurement model and resulting distance metric for each datatype in the study: fluorescence time-course data, synthetic ordinal time-course data, and synthetic survival vs death binary data for a sample of 400 initial conditions. These likelihood functions were used to calibrate the models to each dataset. In addition to their use in the likelihood formulation, the measurement models, were also used to generate synthetic non-quantitative datasets.

**2.1. Calibration to fluorescence data.** We first trained the aEARM to normalized fluorescence time-course data for IC-RP and EC-RP, i.e., fluorescent proxies for substrates of initiator and effector caspase, respectively (i.e., Bid and PARP, respectively). Consistent with previous work, we defined a likelihood that assume an i.i.d. Gaussian-noise component $\epsilon \sim N(0, \sigma^2)$ on normalized tBID and cPARP predictions of the aEARM; where $\sigma^2$ assumedly equals the variance of the data [21,38]. This yields a log-likelihood function (Eq 13) where data the, $\hat{y}$, and normalized aEARM predictions, $y$, are compared for each time-point, $t$, and observable, $i$ (i.e. tBID/IC-RP and cPARP/EC-RP). The aEARM trained to these fluorescence data served as the starting point in the synthesis of ordinal, nominal, mixed, etc. datasets, below.

$$\log \mathcal{L}(\hat{\boldsymbol{y}}|\boldsymbol{\theta}) = \sum_1^N \sum_t^T -1/2\sigma_i(t)^2 \times (\hat{y}_i(t) - y_i(t, \boldsymbol{\theta}))^2 \tag{13}$$

**2.2. Calibration to ordinal data.** To train the aEARM to synthetic ordinal time-course data, a measurement model (i.e., that models the probability of each ordinal category as a function of an aEARM variable) was defined and applied in the formulation of a likelihood function [39]. The ordinal logistic regression python package, MORD, applies empirical ordering constraints to Scikit-Learn's logistic regression class; this class then calculates a probability for each ordinal category [39]. The ordinal logistic model, encoded in MORD, defines ordinal constraints as a linear function of predicted values of an aEARM variable (Eq 14). For aEARM variable, $x_{tBID}$, where each ordinal constraint, $j$, is a logistic function $\varphi(z)$ with a different offset coefficient, $\beta_j$, but shared slope coefficient, $\alpha$, for each of the ordinal categories.

$$p(y_{tBID} \geq c_j | x_{tBID}) = \varphi(\alpha x_{tBID} + \beta_j) \tag{14}$$

We express each offset, $\beta_j$, using a distance, $\theta_j$, from its preceding offset term; $\beta_j = \beta_{j-1} + \theta_j$. The first offset, $\beta_0 = \theta_0$, and subsequent distance, $\theta_j$. To satisfy the empirical ordering constraint for ordinal measurement models, we then ensured the different offsets are

monotonically increasing (i.e., $\beta_j \geq \beta_{j-1}$) by assigning only positive values for the domain of $\theta_j$, and its priors. See *Methods Section 5* for more detail.

We represent each ordinal value using the *cumulative model*. The cumulative model assumes ordinal values $y_i(t)$ originate from an underlying continuous latent variable $x_i(t, \boldsymbol{\theta})$), and the probability of each ordinal value is the probability of the latent variable exceeding a threshold that defines $y_i(t) \geq c_j$ minus the probability of the latent variable exceeding a threshold that defines $y_i(t) \geq c_{j+1}$ (See Eq 15) [40–42].

$$P(y_i(t) = c_j | x_i(t, \boldsymbol{\theta})) = P(y_i(t) \geq c_j | x_i(t, \boldsymbol{\theta}), \alpha_i, \beta_{i,j}) - P(y_i(t) \geq c_{j+1} | x_i(t, \boldsymbol{\theta}), \alpha_i, \beta_{i,j+1}) \quad (15)$$

The underlying continuous latent variable assumption of the cumulative model is appropriate for our study since the latent variable, in our case, is intracellular concentration. These offset and slope coefficients are additional free parameters to be inferred in the model calibration. For example, a measurement model with $J$ categories can be defined using $J-1$ ordinal constraints and will therefore add a total of $J$ free parameters (i.e., $J-1$ offset coefficients and 1 shared slope coefficient) to the model. Alternative ordinal models: sequential models (which reflect successive transitions to higher categories) and adjacent-category models (which reflects comparisons between pairs of adjacent categories and provides useful mathematical properties) may be appropriate in other specific contexts [42].

We also encoded error in our synthetic ordinal data by defining a 5% misclassification probability, i.e. we assume 95% probability the reported ordinal category, $c_j = y$, and 2.5% probability of adjacent categories, $c_j = y\pm1$, (5% for adjacent terminal categories). We model this by the marginal probability that the observation classified into the category predicted by the model: $\sum_j^K P(\hat{y}_i(t) | y_i(t) = c_j)$ [41]. Together, this yields a log-likelihood function (Eq 16) where the probability of each category $c_j$ is calculated for each time-point, $t$, and observable, $i$; and applied toward a likelihood of the data $\hat{y}$ given the model. Where noted, we also trained the aEARM using measurement models with preset fixed parameters (Table C in S1 Text).

$$\log \mathcal{L}(\hat{y} | \boldsymbol{\theta}, \boldsymbol{\alpha}, \boldsymbol{\beta}) = \sum_i^N \sum_t^T \log \sum_j^J P(\hat{y}_i(t) | y_i(t) = c_j) P(y_i(t) = c_j | x_i(t, \boldsymbol{\theta}), \alpha_i, \beta_{i,j}) \quad (16)$$

**2.3. Calibration to nominal data.** We trained aEARM to synthetic binary (survival vs death) data by incorporating a measurement model (i.e., logistic model of the probability of each categorical outcome) like that used for the ordinal data. We used the Scikit-Learn logistic regression class to model the probability of a cell-death outcome, $y = c_1$, as a linear function of features, $x_l$, derived from the aEARM simulation (Eq 17) where $\alpha$ is a slope term, $\beta$ is an intercept and $\beta_l$ are weight coefficients for each of the $L$ features [43].

$$p(y = c_1 | \boldsymbol{x}) = \varphi(\alpha(\beta + \sum_l^L \beta_l x_l)) \quad (17)$$

Previous studies used *a priori* knowledge and assumptions about which features of a cell-fate marker's dynamics to associate with the binary outcome. For instance, recent work delineates necrotic and survival cell fate outcomes using a threshold in the concentration of a known necroptosis marker (this assumption enabled models of necroptosis in the absence of an established relationship between the dynamics of the marker and commitment to necroptosis) [44]. Roux et al, investigated an empirical relationship between initiator caspase reporter protein (IC-RP), a fluorescent indicator of caspase activity or proxy for caspase substrate cleavage, and apoptosis in TRAIL stimulated HeLa cells [20]. They found, instead of concentration, the maximum rate of change in IC-RP and the time when that rate of change maximized better predicted the apoptosis-survival decision [20]. The features we use in our study are based on findings by Roux et al [20]. The features are derived from aEARM simulated tBID dynamics,

$x_{tBID}(t, \boldsymbol{\theta})$: time at maximum rate of change, and log-maximum rate of change. To test the measurement model's ability to discriminate between predictors and non-predictors of cell death, we encoded an additional feature: the concentration of an unrelated non-apoptotic species (USM2 in S1 Data) when bid truncation maximizes. Together this totals three features. We interpret each observation in the dataset as an independent Bernoulli random variable. Each cell death vs survival observation is compared with these three features, $x_{l,m}$, extracted from an aEARM trajectory that was simulated from a unique vector of initial conditions. There were 400 observations; 2 sets of 200 observations corresponding to 10 and 50ng/mL initial ligand concentration. Together, this yields a log-likelihood function (Eq 18) where each, $m$, of the $M$ aEARM simulated trajectories corresponds to an observation $\hat{y}_m$. Given the definitiveness of observed surviving vs dead outcomes, we considered the chance of misclassification to be zero (i.e., $P(\hat{y}_m|y_m = c_1) = 0$ when $\hat{y}_m \neq c_1$).

$$\log \mathcal{L}(\hat{\boldsymbol{y}}|\boldsymbol{\theta}, \boldsymbol{\alpha}, \boldsymbol{\beta}) = \sum_m^M P(\hat{y}_m|y_m = c_1)\log \varphi(\alpha(\beta + \sum_l^L \beta_l x_{l,m})) + \sum_m^M (1 - P(\hat{y}_m|y_m = c_1))\log[1 - \varphi(\alpha(\beta + \sum_l^L \beta_l x_{l,m}))] \quad (18)$$

**2.4. Calibration to cell viability data.** We trained aEARM to published fractional cell death data by Wajant et al [29]. We used the Scikit-Learn Gaussian process regression $\boldsymbol{y}(\boldsymbol{x}) \sim GP(m(\boldsymbol{x}), \kappa(\boldsymbol{x}, \boldsymbol{x}'))$ (i.e., Eq 5) to map variables in aEARM to values of cell viability in the data $\boldsymbol{y}(\boldsymbol{x})$. In this case, $\boldsymbol{x}$ is vector of features extracted from aEARM predicted tBID dynamics (described below) for each experiment represented in the data by Wajant et al [29, 45]. Wajant et al. investigated how diminished FADD expression affected cell viability of HeLa cells treated with different concentrations of TRAIL [29]. The low- and high-dominant negative FADD genotypes of Wajant et al. were modeled via free parameters, $\delta_{low}$ and $\delta_{high}$, which represent diminished rates of DISC formation relative to wildtype $kc_0$ in aEARM. The TRAIL concentrations were encoded in aEARM as different values concentrations of ligand. $L_0$. Therefore, each cell viability value corresponds to a vector of features $x$ extracted an aEARM trajectory that was simulated from a unique experiment-specific vector of initial conditions and rate parameters.

The kernel in our Gaussian process model, $\kappa(\boldsymbol{x}, \boldsymbol{x}')$, is a radial-basis kernel (Eq 20), where $x_p$ and $x_q$ are features corresponding to two points in the data ($\hat{y}_p$ and $\hat{y}_q$) and $\sigma_{\hat{y},p}^2$ is observed standard deviation at $y_p$. A single Gaussian process model was applied to all the points in the cell viability dataset. For any finite set of points (e.g., the dataset by Wajant et al.), this process defines a join Gaussian (Eq 19), where $\boldsymbol{X}$ is the features, $x$, for the entire dataset, $K_{i,j} = \kappa(\boldsymbol{x_i}, \boldsymbol{x_j})$, and $\boldsymbol{\mu} = (m(x_1), m(x_2)\ldots m(x_N))$ is the cell viability values for the entire dataset [44].

$$p(f_M|\boldsymbol{X}) = \mathcal{N}(f_M|\boldsymbol{\mu}, \boldsymbol{K}) \quad (19)$$

The coefficient $\sigma_f^2$ and variance term $M$, (Eq 21) are free parameters that are estimated in the calibration (these are represented as $\theta_M$ in Eq 22). The variance term (Eq 21) is modeled as the diagonal matrix where $\boldsymbol{\beta}$ are the weight coefficients for the features extracted from tBID dynamics predicted by aEARM [45].

The features we use in our model were critical points (i.e., relative maxima and minima, points of inflection) extracted from the aEARM simulations. Specifically, we used second- and fourth-order finite difference approximations of the first and second derivatives [46], respectively, to extract relative maxima, minima, and points of inflection for aEARM simulated tBID trajectories. We considered the time, tBID concentration at the relative maxima and minima; the time, tBID concentration and BID truncation rate at the point of inflection; whether such critical points existed on a trajectory; and the maximum BID truncation rate as potential

features. Different parameterizations of aEARM provide different sets of critical points. To keep a consistent parameterization of the features weights, throughout the model calibration, a look-up table was employed where new critical points were appended to the table as they arose during model calibration. The look up table took a maximum of ten features.

We modeled the likelihood of the data given the model (and measurement model) as the log-marginal likelihood of the Gaussian process model given the data (Eq 22); where the kernel which is a function of the model parameters $\theta$ and the measurement models $\theta_M$ [45].

$$\kappa\left(x_p, x_q\right) = \sigma_f^2 \exp\left(-\frac{1}{2}(x_p + x_q)^T M(x_p + x_q)\right) + \sigma_{y,p}^2 \tag{20}$$

$$M = \mathrm{diag}(\boldsymbol{\beta})^{-2} \tag{21}$$

$$\log \mathcal{L}(\hat{y}|\theta, \theta_M) = -\frac{1}{2}\hat{y}^T(K(\theta_M, \boldsymbol{x}(\theta)) + \sigma_y^2 I)^{-1}\hat{y} - \frac{1}{2}\log|K(\theta_M, \boldsymbol{x}(\theta)) + \sigma_{\hat{y}}^2 I| - \frac{n}{2}\log 2\pi \tag{22}$$

## 3. Generating synthetic datasets

The calibration, by Ortega et al., of aEARM to IC-RP and EC-RP fluorescence time-course data provided an optimally fit vector of rate coefficient parameters, which served as the "ground truth" parameter vector in the synthesis of the nonquantitative datasets (Table E in S1 Text) [23]. These parameters were applied to aEARM, and the resulting aEARM was used simulate time-courses for variables to be indicated in the nonquantitative data: truncated BID (tBID), initiator caspase localization to the death inducing signaling complex (IC-DISC), and cleaved PARP (cPARP).

These time-courses were converted to ordinal time-course datasets. The effective bit resolution of a measurement technology dictates how many unique values it can distinguish [47]. The total number of ordinal categories, *J*, was set such that resulting dataset had less than (but approximately) 70% of the effective bit resolution, *EBR*, (Eq 23) of the IC-RP of EC-RP data. Eqs 24 and 25, and their constants, are standard derivations of the effective resolution of noisy analogue signals [47]. The signal to noise ratio, *SNR*, (Eq 25) assumes the data, *d*, were subject to Gaussian noise and a 0.10 misclassification rate between adjacent values; modeled as the 0.95 quantile of a unit normal distribution [47]. Therefore, the number of ordinal categories were 5 and 4 for tBID and cPARP, respectively. The number of ordinal categories for IC-DISC were arbitrarily set to 4. Arbitrary values of slope and offset coefficients (Table C in S1 Text) were designated "ground truth" and applied to ordinal measurement models (described in *Methods Section 2.2*). The resulting measurement models map the values in the aEARM simulated time-courses to probabilities of each ordinal category. These probabilities were used to simulate random class assignments for synthetic ordinal datasets. The aEARM was trained to time-course ordinal values of tBID and cPARP or time-course ordinal values of IC-DISC and nominal data described below.

$$J \leq 0.7 \times 2^{EBR} \tag{23}$$

$$EBR = (SNR - 1.76)/6.02 \tag{24}$$

$$SNR = 20 \log_{10}((\max d - \min d)/(q_{0.95}\mathrm{rms}(d))) \tag{25}$$

To generate synthetic nominal (binary cell survival vs death) data, two heterogeneous populations of 200 aEARM tBID (and an unrelated non-apoptotic species, USM2) trajectories were simulated from ground truth parameters. The populations had distinct initial ligand

concentrations (10 or 50 ng/mL). Heterogeneity was modeled by a log-normal random sample of certain initial conditions (described in *Methods Section 1*). These time-courses were preprocessed to yield values of the features encoded in nominal measurement model, above. This measurement model (which was encoded with preset "ground truth" values of slope, intercept, and weight coefficients–See Table D S1 Text) maps these features to probabilities of the binary outcomes. These probabilities were used to simulate random class assignments for synthetic nominal datasets (Fig 2B).

To generate a synthetic distribution of times at which Bid truncation was half-maximal, two heterogenous populations of 200 aEARM tBID time-courses, corresponding to 10 and 50ng/mL initial ligand concentrations, were simulated from ground truth parameters (as above). Time at half-maximal tBID was calculated via linear interpolation and rounded to the nearest 3-minute time-point (i.e., to reflect temporal resolution of common time-series intra-cellular experiments) (Fig 2A).

## 4. Fractional cell death dataset

We calibrated the aEARM parameters to published fractional cell death measurements (Fig 5A) taken Wajant et al via visual inspection of the published figure [29]. These measurements were taken of HeLa cell in which apoptosis was inhibited through graded expression of a dominant negative FADD. A test set of fractional cell death data was taken from Orzechowska, et al.,) via visual inspection of the published figure [30]. The test data were measurement of HeLa cells in which Bid expression was overexpressed (i.e., TAT Bid HeLa cells). Bid expression in TAT Bid HeLa was modeled as 20% increased Bid (which is consistent with the observations of Orzechowska, et al.) [30].

## 5. Model Calibration via Bayesian Inference

The aEARM was calibrated using DREAM(ZS) algorithm for all datasets [48]. Rate parameters in aEARM were given independent log-normal prior probability functions with a location equal to the ground-truth parameter vector (i.e., the maximum a posteriori parameter vector extracted from the aEARM calibration by Ortega et al.) and a scale term of 1.5.

Wildtype, low- and high-dominant negative FADD genotypes were modeled via free parameters, $\delta_{low}$ and $\delta_{high}$, which represent diminished rates of DISC formation relative to wildtype $kc_0$ in aEARM. These parameters, $\delta_{low}$ and $\delta_{high}$, were given independent truncated log-normal prior probability function with locations equal to 0 and -1 respectively. The prior probability functions were truncated at 0 and negative infinity. The nominal (cell death vs survival) dataset features a heterogeneous population of values. We modeled this heterogeneity with a random sample of initial conditions (described above). This random sample was shifted and scaled according to inferred values of the model mean and variance. The mean (if estimated) was given a log-normal distribution prior probability function with a location equal to ground-truth and a scale term of 1.5. The extrinsic noise (or variance) was given inverse gamma distribution with $a$ and $b$ terms such that the resulting coefficient of variation had a prior mean and standard deviation of 0.20 and 0.015 respectively.

Prior probability functions were also applied to the measurement models' free parameters. To encode empirical ordering constraints on the ordinal measurement model (Eq 14.), the slope terms, $\alpha$, were greater than zero; they were given independent exponential distribution prior probability functions (with location of 0.0 and scale of 100.0). To insure monotonically increasing offset terms, each offset, $\beta_j$, was defined by the distance, $\theta_j$, from its preceding offset term; $\beta_j = \beta_{j-1} + \theta_j$. The first offset, $\beta_0 = \theta_0$, and subsequent distance, $\theta_j$, terms were given independent exponential distribution prior probability functions (with location of 0.0 and scale of 0.25). We

explored the effect of increasingly biased priors on the ordinal measurement model parameters. Where noted, the slope terms, $\alpha$, were given increasingly constrained independent prior probability functions: uniform (0.0–100.0 bounds), Cauchy (50.0 location and 10.0 scale) and Cauchy (50.0 location and 1.0 scale). The offset, $\beta_0$, and distance, $\theta_j$, terms were similarly given independent uniform (0.0–1.0 bounds), Cauchy (0.2 location and 0.05 scale) and Cauchy (0.2 location and 0.005 scale) distribution prior probability functions. Parameters for the measurement model were given independent Laplace distribution prior probability functions with a location of 0.0 and scale of 1.0 for the slope, $\alpha$, and 0.10 for the intercept and weighting coefficients, $\beta$ and $\beta_l$.

The likelihood functions were described in *Methods Section 2*. Additional settings applied to the DREAM(ZS) algorithm were as follows: number of chains (4) number of crossover points (nCR = 25), adaptive gamma (TRUE), probability of gamma = 1 (p_gamma_unity = 0.10), gamma term resolution (gamma_levels = 8). A burn-in period wherein crossover weights are adapted was set to 50,000-step burn-in for ordinal datasets and 100,000+ step burn-in. The calibration algorithm continued until it reached the stopping criterion: when the Gelman-Rubin metric (calculated on the latter 50% of the traces) was less than or equal to 1.2 for all free-parameters in the model; at which point the parameter traces were considered converged[47]. Gelman-Rubin metrics for each calibration are listed in Table F in S1 Text. Model calibrations were run on a x64 Intel with 32 total CPU threads (256GB RAM) and x64 AMD with 256 threads (1024GB RAM). Run times varied widely given the stochastic nature of the optimization algorithm but were typically one to seven days for simple model calibrations. Random samples of 1000 parameter were taken from the latter 50% of the resulting parameter-traces were used in subsequent analyses. Source code for the model calibrations as well as code for downloading the resulting parameter-traces is found at https://github.com/LoLab-MSM/Opt2Q and https://github.com/LoLab-MSM/aEARM_cell_viability_calibration; data in Table F in S1 Text can be loaded from https://doi.org/10.5281/zenodo.7007655.

## 6. Model predictions

We simulated the equal-tailed 95% credible region of the posterior predictions of aEARM via samples of the model parameters posterior distribution. This was done by randomly generating 1000 parameter sets sub-sampled from the posterior sample of parameters generated via PyDREAM. For each parameter set, tBID time-courses (and/or cPARP, IC-DISC) were simulated from aEARM. The 95% credible region of the predictions was then determined via 0.025 and 0.975 quantile bounds on the tBID (or other variables) values for each time-point in the simulated time-course. The area bounded in the 95% posterior credible interval was determined by summing the difference between the 0.025 and 0.975 quantile bounds across 100 equally spaced time points on the trajectory. The 95% posterior credible intervals on the measurement model predictions were similarly the described by calculating 0.025 and 0.975 quantile boundaries on the predictions of the measurement model parameterized via 1000 parameter set samples from a posterior. This includes the posterior probability distributions of the feature coefficients encoded in the nominal measurement model. To model predictions of the nominal dataset, however, we randomly generated 100 parameter sets via sub-sampling of the posterior parameter distribution. For each parameter set, we simulate tBID dynamics from the set of 400 initial conditions as described above; from that we compute maximum BID truncation rate and time at maximum BID truncation rate for each of the 400 trajectories. The 0.05 contour of the KDE of the resulting 400 values of maximum BID truncation rate and time at maximum BID truncation rate was plotted for each of the 100 parameter sets.

We also simulated posterior predictions fractional cell death. This was done by randomly generating 1000 parameters sets subsampled from the posterior sample of parameters

generated via PyDREAM. For each parameter set tBID time-courses were simulated from aEARM and preprocessed to extract a critical points representation of the tBID dynamics. These features were supplied to the parameterized Gaussian process model of our fractional cell death measurements, which returned a prediction for the mean fractional cell death. Violin plots of the KDE of the resulting sample of 1000 posterior predictions of fractional cell death were plotted in Fig 5.

## Supporting information

**S1 Text. Supporting Information file containing Tables A-E and Figs A-T.** The supporting information provides method details. Tables A-E describe parameters and reactions included in the aEARM. The following figures describe the prior and posterior distributions of aEARM parameters calibrated to using different dataset and measurement models. Also included (Figs A, G, K and R) are various posterior predictions for aEARM variables. This file contains a table of contents for further details.
(DOCX)

**S1 Data. Gelman-Rubin Convergence Statistics for each aEARM calibration.** The Gelman-Rubin convergence statistics for the aEARM parameters for each calibration scenario were tabulated. This table can be loaded from https://doi.org/10.5281/zenodo.7007655.
(XLSX)

## Acknowledgments

The authors would like to thank Dr. Alexander Lubbock, Dr. Leonard Harris, and Dr. Vito Quaranta, for insightful conversations and critical feedback on this work.

## Author Contributions

**Conceptualization:** Michael W. Irvin, Arvind Ramanathan, Carlos F. Lopez.

**Formal analysis:** Michael W. Irvin.

**Funding acquisition:** Carlos F. Lopez.

**Investigation:** Michael W. Irvin, Carlos F. Lopez.

**Methodology:** Michael W. Irvin, Arvind Ramanathan.

**Project administration:** Carlos F. Lopez.

**Software:** Michael W. Irvin, Arvind Ramanathan.

**Supervision:** Carlos F. Lopez.

**Validation:** Michael W. Irvin.

**Visualization:** Michael W. Irvin.

**Writing – original draft:** Michael W. Irvin, Carlos F. Lopez.

**Writing – review & editing:** Michael W. Irvin, Arvind Ramanathan, Carlos F. Lopez.

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
