## [Decision Letter · Decision Letter 0]

18 Jun 2022

Dear Dr. Lopez,

Thank you very much for submitting your manuscript "Predictive uncertainty in mechanistic models of cellular processes calibrated to experimental data" for consideration at PLOS Computational Biology.

As with all papers reviewed by the journal, your manuscript was reviewed by members of the editorial board and by several independent reviewers. In light of the reviews (below this email), we would like to invite the resubmission of a significantly-revised version that takes into account the reviewers' comments.

As you will see, one reviewer, in particular, had very strong concerns regarding a lack of diligence in the preparation of the manuscript. It would be important to address these concerns.

We cannot make any decision about publication until we have seen the revised manuscript and your response to the reviewers' comments. Your revised manuscript is also likely to be sent to reviewers for further evaluation.

Sincerely,

Martin Meier-Schellersheim

Associate Editor

PLOS Computational Biology

Jason Haugh

Deputy Editor

PLOS Computational Biology

Reviewer's Responses to Questions

**Comments to the Authors:**

Reviewer #1: The authors make two significant contributions in my estimation. First, they extend the Bayesian inference methodological concepts of Ref. 12, wherein categorical observations of continuous variables are considered for use in model parameterization. In that work, the categorical observations are viewed as arising from a classifier function which has known threshold parameters. In this manuscript, the authors generalize, considering other types of functional relationships (GP/logistic) between observations and model/process variables and considering a wider range of combinations of observational data types. The authors argue that the function relating observations and model/process variables can be anything, such as a universal function approximator, and that the function may have different kinds of parameters besides thresholds, which can be learned jointly with model parameters. I think these are important ideas. The second contribution is that the authors provide another demonstration of how non-quantitative observations can be used to estimate model parameter values. There is some earlier work in this area but not much and the new demonstration differs from those provided earlier in several ways. The weakness of the demonstration is that it relies on synthetic data but this is offset by using an empirically-calibrated model to generate the synthetic datasets.

Minor:

In the second paragraph of the Introduction, when discussing earlier work, it might be an improvement to include a clearer objective summary of earlier approaches and only then to present an evaluation of their limitations. It seems that the authors are criticizing the methodology of Ref. 12 (closest to the authors’ own work) because in that work thresholds are taken to be known. This is an assumption that can easily be relaxed in principle, so it is not a severe limitation. It is certainly fair to say that taking a threshold to be known can introduce bias, but it should probably be mentioned that taking some parameters to be fixed is commonly done in inference problems to make the inference problem practically solvable. It is stated that other earlier work also has limitations but these limitations are somewhat unclear to me from just reading the text. The limitations of all earlier methods discussed should be clarified, perhaps through the discussion of concrete examples.

Consider defining the following terms in the text upon first use: nominal (binary categories?), ordinal (multiple categories?), semi-quantitative (e.g., relative measurements?), and quantitative. The discussion in the first paragraph of the Results section about the merits of the different data types could be made more interesting by saying more about the data. The discussion currently focuses on the value of the data, without saying much about the data.

The concepts of “measurand” and “measurement” should be more clearly defined when these terms are first brought up.

The comment “we introduce a concept from statistics, and social sciences: the measurement model” is a big overstatement. Measurement models, such as Eq (6) in Box 1, are commonly used by biological modelers. The overstatement is repeated in the Discussion: “Our work introduces the concept of measurement models to the mechanistic modeling paradigm.” I think what the authors are doing is using the well-known measurement model concept to connect non-quantitative observations to model/process variables, which is innovative. The authors should rephrase for precision and to avoid overstatement.

I would like to see more information about the MCMC sampling, such as representative parameter and likelihood trace plots and pairs plots.

I am concerned about a sentence in the Abstract: “We find two orders of magnitude more ordinal (e.g. immunoblot) data are necessary to achieve accuracy comparable to quantitative (e.g. fluorescence) data.” It should probably be mentioned that this finding is likely to be problem-specific. The authors really only consider one demonstration problem. Consideration of many problems would be needed to start making general conclusions.

Reviewer #2: This paper adds to the growing but small community of modelers who are focused on the role of qualitative data as well as more model directed experimentation, something one might term model driven rather than a data driven approach. A model driven approach is I feel exactly the right approach. The current paper adds to the literature by conducting a more thorough analysis of the role of qualitative data. I was very interested to see a discussion of the concept of measurement model. The theory of development models and the role of data is not discussed very much, if at all in systems biology, and if anything, this paper will introduce the community to this more nuanced and important concept.

Overall, this was an interesting paper, its not a definitive analysis (as the authors admit) but it would be a useful addition to the literature and will hopefully stimulate further similar studies.

Minor:

Line 105. I would add some brief definitions of nominal, ordinal etc. to the text. I understand that Figure 1 describes these terms, but the caption is lengthy. Maybe something along the lines:

“nominal (data is categorized), ordinal (data is categorized and ranked),”. For semi-quantitative and quantitative, I’m don’t quite understand the difference. I would attempt to clarify these two terms more, but I recommend being succinct. Is semi-quantitative unit less data, e.g ratios and quantitative data with units? I wasn’t sure from the text.

Line 144: “The model was calibrated to above fluorescence data”, missing word somewhere?

Line 249: “As described in Methods, we added a synthetic 249 dataset containing 61 ordinal time-course measurements” I didn’t immediately quite understand how time-course measurements can be an ordinal data type (this is briefly explained in the methods section). Could the authors point to the method section (line 631 in the manuscript I assume?) where this is described, for example: As described in the Methods x.y, we added a synthetic…”. The authors indicate in the methods section that: “These time-courses were converted to ordinal time-course datasets”, from reading the text I assume the time course data was just turned into a set of ranked data but without a quantitative time dimension?

I recommend that wherever the main text says “See Methods” or similar, I would be explicit about where in the methods section the reader to go to, this would greatly help the reader.

Line 269-270: “but this time we replaced the 270 free parameters in the measurement model fixed a priori parameterizations”, text doesn’t read correctly? Now sure what is being said here.

Conclusion: I would include in the conclusion a series of bullets points outlining the basic recommendations and results obtained from the work. I think many readers might find it difficult to extract the key conclusions as the text is fairly dense. I think there are perhaps three primary conclusions from the work, these should be corrected or reworded if I am inaccurate here: 1) A lot more ordinal data is required to constrain a model; 2) Prior assumptions about parameter values can bias a model; 3) Use the model to devise the most profitable data to collect.

Line 766: Typo in “We Consider”, Figure 1

Last but not least, the GitHub link for the osurce code appears to take one to a 402 page.

Reviewer #3: review is uploaded as an attachment

**Have the authors made all data and (if applicable) computational code underlying the findings in their manuscript fully available?**

Reviewer #1: Yes

Reviewer #2: **No: **GitHub link appears to be broken.

Reviewer #3: None

PLOS authors have the option to publish the peer review history of their article (what does this mean?). If published, this will include your full peer review and any attached files.

Reviewer #1: No

Reviewer #2: No

Reviewer #3: No
---

## [Decision Letter · Decision Letter 1]

30 Dec 2022

Dear Dr. Lopez,

Thank you very much for submitting your manuscript "Model certainty in cellular network-driven processes with missing data." for consideration at PLOS Computational Biology.

As with all papers reviewed by the journal, your manuscript was reviewed by members of the editorial board and by several independent reviewers. In particular, in light of the review comments provided by reviewer #3, we would like to invite the resubmission of a significantly-revised version that takes into account that reviewer's comments.

Please note that many of the issues mentioned by reviewer #3 should have been addressed in the first revision you submitted to us and that, in general, it is part of the responsibilities of the authors to minimize errors that can be easily detected when proofreading the manuscript. 

We cannot make any decision about publication until we have seen the revised manuscript and your response to the reviewers' comments. Your revised manuscript is also likely to be sent back to reviewer #3 for further evaluation.

Sincerely,

Martin Meier-Schellersheim

Academic Editor

PLOS Computational Biology

Jason Haugh

Section Editor

PLOS Computational Biology

Reviewer's Responses to Questions

**Comments to the Authors:**

Reviewer #1: This manuscript is interesting. Revisions made to address my concerns about the original version are deemed to be adequate.

Reviewer #2: The authors have address my main concerns. I have no further comments. Should be a useful contribution to literature.

Reviewer #3: Review is uploaded as an attachment.

**Have the authors made all data and (if applicable) computational code underlying the findings in their manuscript fully available?**

Reviewer #1: Yes

Reviewer #2: Yes

Reviewer #3: Yes

PLOS authors have the option to publish the peer review history of their article (what does this mean?). If published, this will include your full peer review and any attached files.

Reviewer #1: No

Reviewer #2: **Yes: **Herbert M Sauro

Reviewer #3: No
---

## [Decision Letter · Decision Letter 2]

6 Mar 2023

Dear Dr. Lopez,

We are pleased to inform you that your manuscript 'Model certainty in cellular network-driven processes with missing data.' has been provisionally accepted for publication in PLOS Computational Biology.

Please make sure to correct the issue commented on by the reviewer (below).

Best regards,

Martin Meier-Schellersheim

Academic Editor

PLOS Computational Biology

Jason Haugh

Section Editor

PLOS Computational Biology

Reviewer's Responses to Questions

**Comments to the Authors:**

Reviewer #3: line 446: Eq. 21 should be Eq. 22

**Have the authors made all data and (if applicable) computational code underlying the findings in their manuscript fully available?**

Reviewer #3: None

PLOS authors have the option to publish the peer review history of their article (what does this mean?). If published, this will include your full peer review and any attached files.

Reviewer #3: No

---

## [Editor Report · Acceptance letter]

13 Apr 2023

PCOMPBIOL-D-22-00619R2 

Model certainty in cellular network-driven processes with missing data.

Dear Dr Lopez,

I am pleased to inform you that your manuscript has been formally accepted for publication in PLOS Computational Biology. Your manuscript is now with our production department and you will be notified of the publication date in due course.

With kind regards,

Timea Kemeri-Szekernyes
